# Structure-guided engineering of biased-agonism in the human niacin receptor via single amino acid substitution

Manish K. Yadav[1,3], Parishmita Sarma[1,3], Jagannath Maharana[1], Manisankar Ganguly[1], Sudha Mishra[1], Nashrah Zaidi[1], Annu Dalal[1], Vinay Singh[1], Sayantan Saha [1], Gargi Mahajan[1], Saloni Sharma[1], Mohamed Chami[2], Ramanuj Banerjee [1]✉ & Arun K. Shukla [1]✉

The Hydroxycarboxylic acid receptor 2 (HCA2), also known as the niacin receptor or GPR109A, is a prototypical GPCR that plays a central role in the inhibition of lipolytic and atherogenic activities. Its activation also results in vasodilation that is linked to the side-effect of flushing associated with dyslipidemia drugs such as niacin. GPR109A continues to be a target for developing potential therapeutics in dyslipidemia with minimized flushing response. Here, we present cryo-EM structures of the GPR109A in complex with dyslipidemia drugs, niacin or acipimox, non-flushing agonists, MK6892 or GSK256073, and recently approved psoriasis drug, monomethyl fumarate (MMF). These structures elucidate the binding mechanism of agonists, molecular basis of receptor activation, and insights into biased signaling elicited by some of the agonists. The structural framework also allows us to engineer receptor mutants that exhibit G-protein signaling bias, and therefore, our study may help in structure-guided drug discovery efforts targeting this receptor.

The Hydroxycarboxylic acid receptor 2 (HCA2), also known as the niacin receptor or GPR109A, belongs to the superfamily of G protein-coupled receptors (GPCRs), and it is expressed primarily in the adipose tissues[1–3], keratinocytes[4], immune cells such as neutrophils and Langerhans cells[5] in our body. Upon activation by agonists, GPR109A couples to Gαi sub-family of heterotrimeric G-proteins leading to lowering of cAMP response[6–9]. In addition, activated GPR109A also recruits β-arrestins[10], which are multifunctional proteins involved in GPCR desensitization, trafficking and downstream signaling[11]. Interestingly, GPR109A was identified as the molecular target for the action of nicotinic acid (aka, niacin or Vitamin B3), an effective drug prescribed for lowering the triglycerides, almost two decades ago[12]. Moreover, GPR109A activation also mediates the lowering of LDL (aka, bad cholesterol), enhancing the levels of HDL (aka, good cholesterol)[13]. Furthermore, monomethyl fumarate (MMF), the active metabolite of a psoriasis drug, Fumaderm, and also a therapeutic agent for the

treatment of relapsing forms of multiple sclerosis, has also been identified as an agonist of GPR109A[14–16]. However, activation of GPR109A is also responsible for driving the troublesome side effect of flushing response associated with niacin, acipimox and MMF[10,13,16–19]. This represents a potential limitation with their therapeutic usage, and therefore, additional small molecule agonists targeting GPR109A remains a key focus area[7,19–21].

Several non-flushing agonists, such as MK6892 and GSK256073 with high affinity for GPR109A have been developed and characterized using in-vitro and animal studies although none of these compounds are yet approved for clinical usage[22,23]. In addition, a comprehensive study has also demonstrated that the side effect of niacin-induced flushing response in mouse is driven primarily by β-arrestin-mediated downstream signaling, and therefore, G-protein-biased agonists of GPR109A may represent improved therapeutics compared to niacin[10] (Fig. 1a). In the same study, a previously developed agonist MK0354

[1]Department of Biological Sciences and Bioengineering, Indian Institute of Technology, Kanpur 08016, India. [2]BioEM Lab, Biozentrum, Universität Basel, Basel, Switzerland. [3]These authors contributed equally: Manish K. Yadav, Parishmita Sarma. ✉e-mail: ramanujb@iitk.ac.in; arshukla@iitk.ac.in

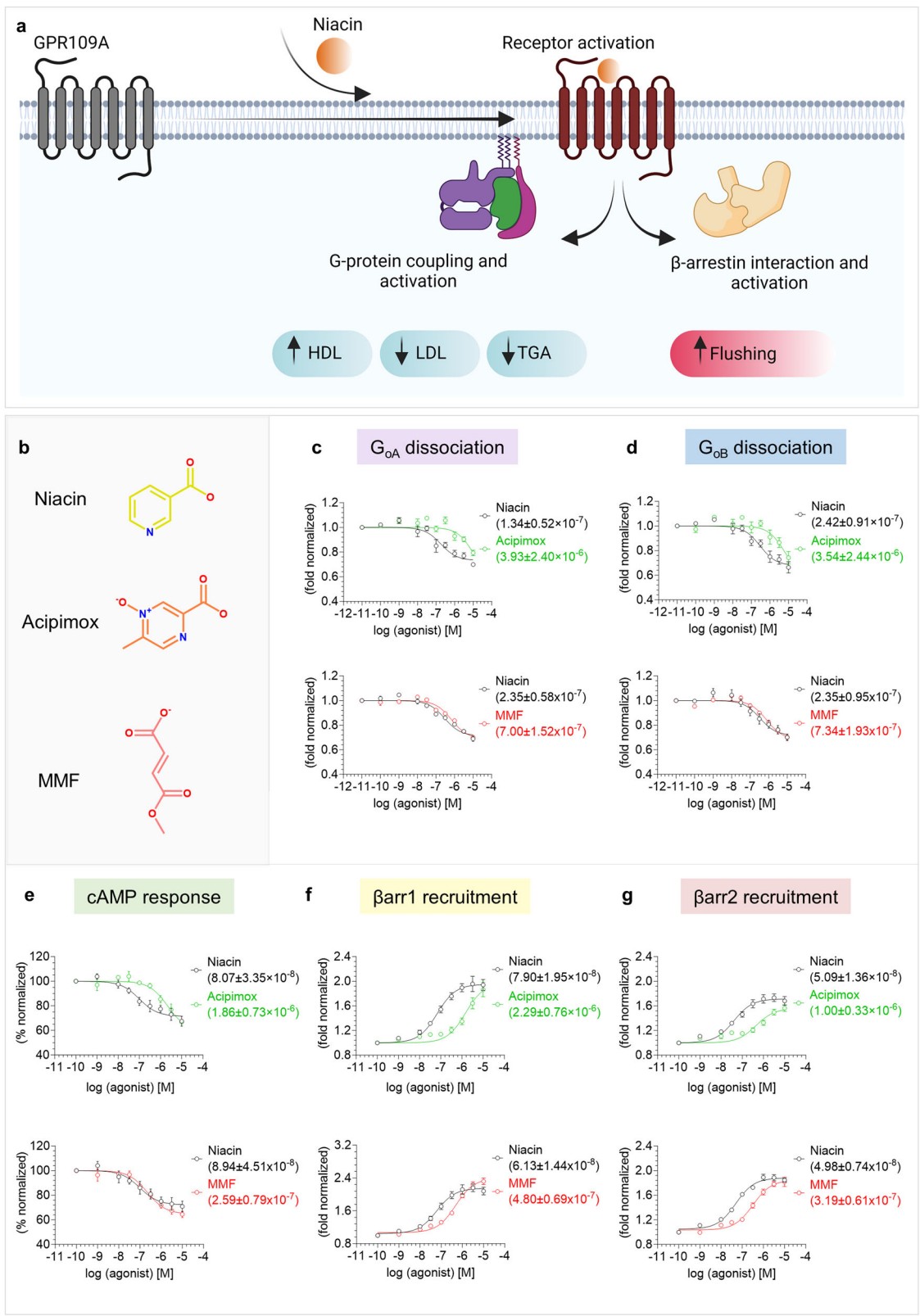

was reported to maintain the anti-lipolytic effect with significant reduction in flushing response, and it was further characterized as a G-protein-biased agonist[10]. Still however, direct structural visualization and molecular mechanism of agonist-binding and activation of GPR109A remain primarily elusive and represent an important knowledge gap to efficiently target this receptor for therapeutic benefits.

Here, we present five different cryo-EM structures of GPR109A-G-protein complexes where the receptor is activated either by niacin, acipimox, MK6892, GSK256073, or MMF. Comparison of these structural snapshots provides the molecular basis of ligand recognition, activation, and transducer-coupling by GPR109A. Importantly, the structural insights allow us to rationally design receptor mutants harboring single amino acid substitution that either renders the receptor

**Fig. 1 | Pharmacological profiling of niacin, acipimox, and MMF on GPR109A.**
**a** Diagrammatic illustration of GPR109A activation and downstream signaling outcomes (created with BioRender.com). **b** Chemical structures of niacin, acipimox, and MMF. **c, d** NanoBiT-based heterotrimeric $G_{oA}/G_{oB}$ dissociation assay in response to acipimox and MMF with niacin as a reference ligand, (mean ± SEM; $n = 3–4$ independent experiments, i.e., for $G_{oA}$ dissociation with acipimox: $n = 4$, for $G_{oA}$ dissociation with MMF: $n = 3$, for $G_{oB}$ dissociation in response to acipimox and MMF: $n = 4$; fold normalized with the minimum concentration for each ligand as 1). **e** Acipimox and MMF stimulated decrease in forskolin-induced cAMP level measured by GloSensor assay (mean ± SEM; $n = 3–4$ independent experiments, i.e, for acipimox induced response: $n = 3$ and for MMF induced response: $n = 4$; % normalized with the minimum concentration for each ligand as 100). **f, g** NanoBiT-based βarr1/2 recruitment in response to acipimox and MMF (mean ± SEM; $n = 4–5$ independent experiments, i.e., for βarr1 recruitment: $n = 4$, for βarr2 recruitment in response to acipimox: $n = 5$ and in response to MMF: $n = 4$; fold normalized with the minimum concentration for each ligand as 1). Source data is provided as Source Data file.

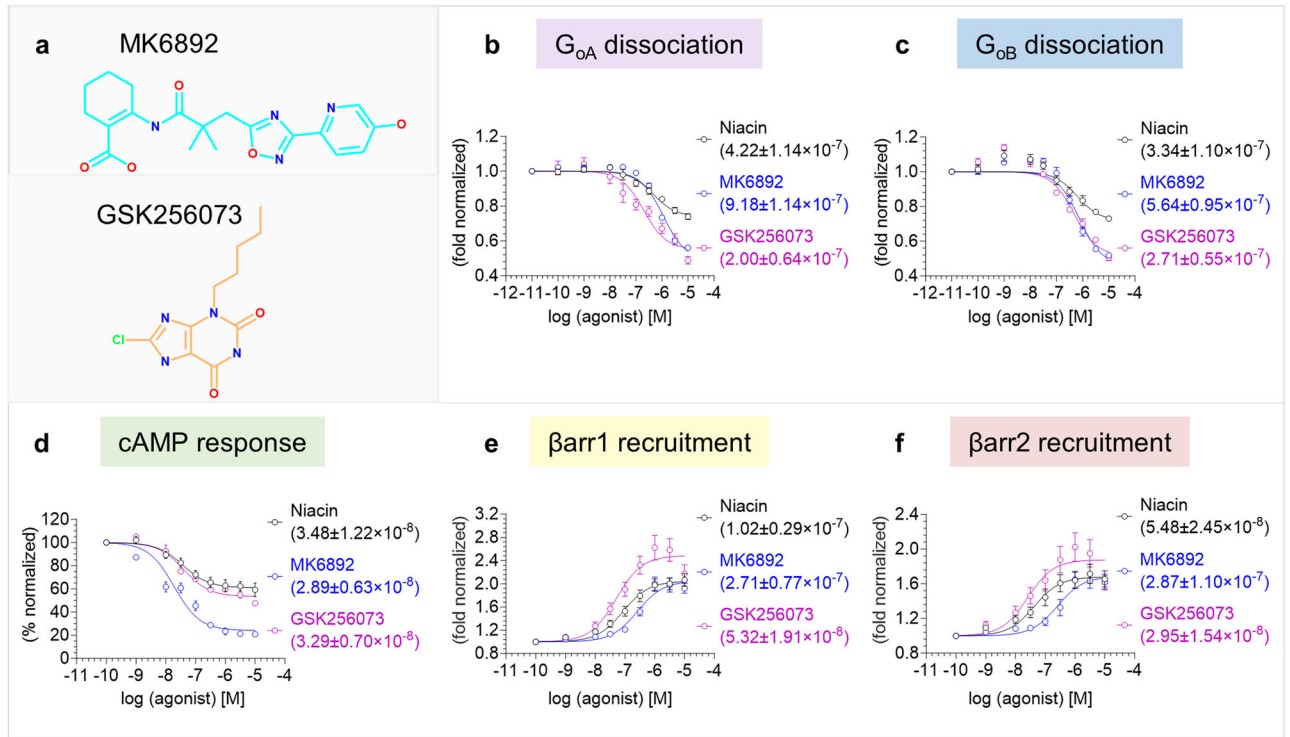

**Fig. 2 | Pharmacological profiling of niacin, MK6892 and GSK256073 on GPR109A.** **a** Chemical structure of MK6892 and GSK256073. NanoBiT-based heterotrimeric $G_{oA}/G_{oB}$ dissociation assay in response to MK6892 and GSK256073 with niacin as a reference ligand. **b, c** showing $G_{oA}$ and $G_{oB}$ dissociation respectively (mean ± SEM; $n = 4$; fold normalized with the minimum concentration for each ligand as 1). **d** Agonist stimulated forskolin-induced cAMP decrease measured by GloSensor assay (mean ± SEM; $n = 4$; % normalized with the minimum concentration for each ligand as 100). **e, f** βarr1/2 recruitment downstream of GPR109A in response to MK6892 and GSK256073 was studied by NanoBiT-based assay (mean ± SEM; $n = 6$; fold normalized with the minimum concentration for each ligand as 1). Source data is provided as Source Data file.

completely inactive with respect to transducer-coupling or impart significant transducer-coupling bias. Our study not only illuminates the structural pharmacology of GPR109A ligands and paves the way for structure-guided discovery of potential therapeutics but also offers a framework to leverage the structural information to rationally encode signaling-bias in GPCRs.

## Results

### Pharmacology of GPR109A agonists used for structural analysis
In order to visualize the molecular framework of ligand recognition and receptor activation, we selected five different ligands namely, niacin, acipimox, MMF, MK6892, and GSK256073 (Figs. 1b, 2a). Of these, niacin and acipimox are clinically prescribed drugs to treat dyslipidemia, while MK6892 and GSK256073 have been developed as non-flushing agonists of GPR109A. MK6892 is a biaryl cyclohexene carboxylic acid derivative that was reported to exhibit high affinity for GPR109A without significant off-target profile, and also displayed reduced vasodilation in animal studies while maintaining free fatty acid reduction similar to niacin[22]. GSK256073 was reported to display robust specificity for GPR109A over the other hydroxycarboxylic acid receptor subtypes, maintain the ability to lower the levels of non-

esterified fatty acids in pre-clinical animal studies with reduced flushing response, and even promising outcomes in healthy male subjects[23]. MMF on the other hand, is the active metabolite of psoriasis drug, Fumaderm, and is also used as a therapeutic agent in multiple sclerosis[14,15,24]. Our selection of these ligands was based on their diverse chemical structures, therapeutic profile, and associated side effects with the goal to understand their interaction with GPR109A and potentially link the structural insights with their therapeutic profile.

We measured the pharmacological profile of acipimox, MMF, MK6892, and GSK256073 with niacin as a reference agonist of GPR109A in G-protein response and βarr (β-arrestin) recruitment assays (Figs. 1c–g, 2b–f). In all these cellular assays, the surface expression of GPR109A was measured using a previously described whole cell-based ELISA method with mock-transfected cells as negative control (Supplementary Fig. 1). We observed that acipimox behaved as a full agonist but with lower potency in G-protein dissociation, cAMP response, and βarr recruitment assay. On the other hand, MMF behaved as a full agonist in both G-protein and βarr recruitment assay with similar efficacy as niacin but weaker potency in βarr assays (Fig. 1c–g, Supplementary Fig. 2a). Moreover, MK6892 exhibited higher efficacy in G-protein response and similar efficacy but

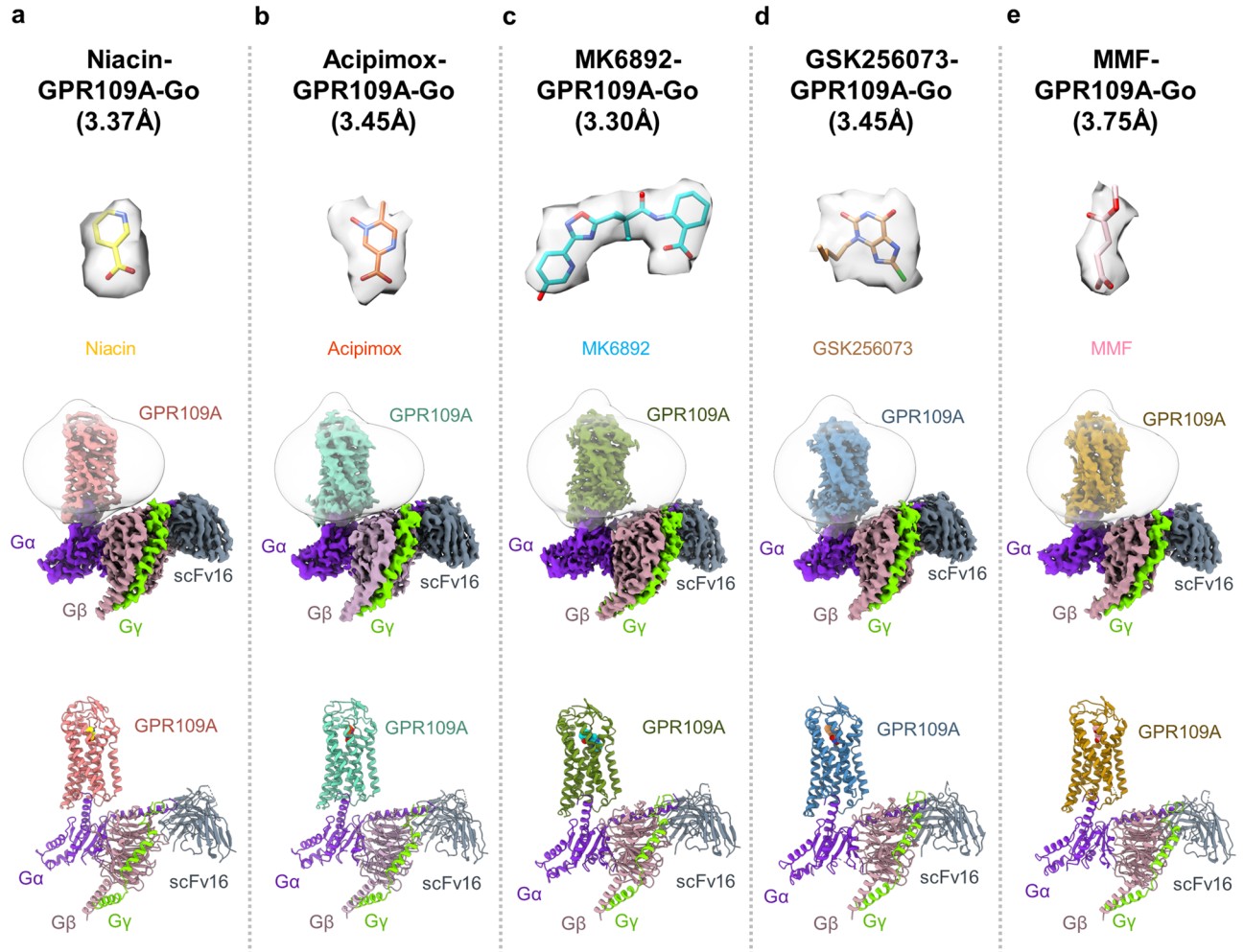

**Fig. 3 | Overall architecture of niacin, acipimox, MK6892, GSK256073 and MMF bound GPR109A-G protein complexes.** Map and ribbon diagram of the ligand-bound GPR109A-Go complexes (front view) and the cryo-EM densities of the ligands (sticks) are depicted as transparent surface representations. **a** niacin-GPR109A-Go: Light coral: GPR109A, blue violet: miniGαo, rosy brown: Gβ1, chartreuse: Gγ2, gray: ScFv16, **b** acipimox-GPR109A-Go: medium aquamarine: GPR109A, blue violet: miniGαo, rosy brown: Gβ1, chartreuse: Gγ2, gray: ScFv16, **c** MK6892-GPR109A-Go: olive drab: GPR109A, blue violet: miniGαo, rosy brown: Gβ1, chartreuse: Gγ2, gray: ScFv16, **d** GSK256073-GPR109A-Go: Steel blue: GPR109A, blue violet: miniGαo, rosy brown: Gβ1, chartreuse: Gγ2, gray: ScFv16, **e** MMF-GPR109A-Go: Dark golden rod: GPR109A, blue violet: miniGαo, rosy brown: Gβ1, chartreuse: Gγ2, gray: ScFv16.

lower potency in βarr recruitment compared to niacin (Fig. 2b–f, supplementary Fig. 2a). In the case of GSK256073, we observed a higher response in G-protein dissociation but similar efficacy in cAMP response, and it also displayed higher efficacy in βarr recruitment as compared to niacin (Fig. 2b–f, Supplementary Fig. 2a). Analysis of these pharmacology data and calculation of the bias factor suggest that MK6892 and MMF act as G-protein biased agonist at GPR109A (Supplementary Fig. 2b, c).

## Overall structure of agonist-bound GPR109A-G-protein complexes

We reconstituted the agonist-GPR109A-G-protein complexes using purified components following previously described methodology successfully applied to other GPCR-G-protein complexes[25]. We determined the cryo-EM structures of these complexes at an estimated resolutions of 3.37 Å, 3.45 Å, 3.3 Å, 3.45 Å and 3.75 Å respectively, for the niacin, acipimox, MK6892, GSK256073 and MMF-bound receptor (Fig. 3a–e, Supplementary Figs. 3–11). The unambiguous densities of the cryo-EM maps enabled us to assign nearly the entire transmembrane domain of the receptor although the first seven residues at the N-terminus of the receptor and the last sixty residues at the carboxyl-terminus were not resolved in the structures potentially due to their

inherent flexibility (Supplementary Figs. 11–13). In addition, in each of these complexes, the ligand densities were reasonably clear to allow us to visualize ligand-receptor interactions, and the map quality at the receptor-G-protein interface facilitated the identification of residue level interactions driving G-protein coupling to the receptor (Supplementary Fig. 12). The precise sequence of the components resolved in these structures is listed in Supplementary Fig. 13. The overall structures of GPR109A in all five complexes are highly similar with an RMSD of 0.6–1.0 Å[2] along the Cα of the receptor interface (Fig. 4) and the key differences are observed in the ligand-receptor interaction as outlined in the sections below.

Structural analysis of the receptor component in these structures uncovered several interesting features. For example, the N-terminus of the receptor in all five structures adopts a twisted β-hairpin structure that positions itself above the extracellular opening of the receptor (Fig. 4a). Moreover, the I169-L176 segment in the ECL2 adopts a twisted antiparallel β-hairpin conformation while S178[4.51]-S181[5.31] dips down into the core of the TM bundle to form part of the orthosteric binding pocket (Fig. 4a). Interestingly, the ECL2 hairpin interacts with the β-hairpin formed by the N-terminal residues L11-C18 to form a "lid-like" architecture that covers the extracellular opening of the receptor (Fig. 4a). This observation can be

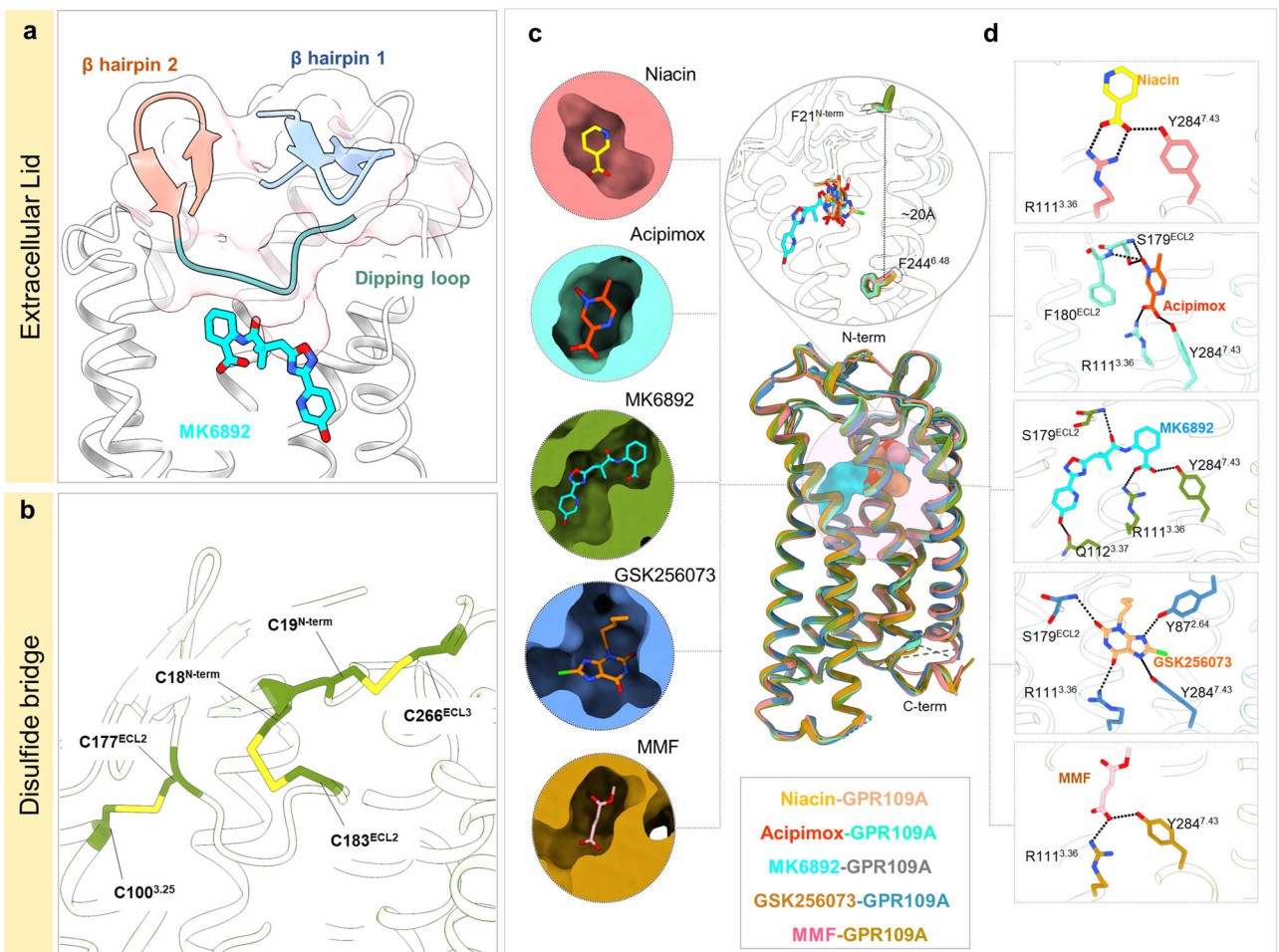

**Fig. 4 | Structural features of ligand binding pocket of GPR109A. a, b** Structural features of ligand (MK6892) bound-GPR109A, N-terminal β-hairpin, close-up view of ECL2 dipping into the orthosteric pocket and ribbon diagram of disulfide bridges formed at the ligand binding pocket. **c** Superposed niacin, acipimox, MK6892, GSK256073 and MMF bound GPR109A structures highlighting the orthosteric binding pocket (cross-sections of GPR109A bound to the individual ligand). **d** GPR109A ligand binding pocket highlighting the major interactions of the individual ligand (black dotted line represents H-bond and ionic interactions).

attributed to the presence of three disulfide bridges (N-terminus C18 with C183 of ECL2, N-terminus C19 with C266 of ECL3, and C177 of ECL2 with C100 of TM3) which helps to further stabilize the N-terminus-ECL2 lid (Fig. 4b). These disulfide bonds might impose additional constraints towards the flexibility of the lid and facilitate docking of the ligand within the orthosteric pocket of the receptor. It is interesting to note that a similar "lid-like" conformation adopted by the N-terminus and ECL2 has been previously reported for several GPCRs such as rhodopsin and CXCR4[26,27], removal of the disulfide constraints results in either complete loss or decreased agonist affinity[28].

**Agonist-receptor interaction in the orthosteric binding pocket**

The ligand binding site in GPR109A is positioned approximately 20 Å deep in the receptor core (measured from N-terminal F21 to F244[6.48]) (Fig. 4c), and all five agonists share a common interaction interface, at least in part, on the receptor where chemically similar moieties of the ligands are positioned (Fig. 4c). An array of aromatic residues namely, F276[7.35], F277[7.36], W91[ECL1], F180[ECL2] and F193[5.43]; and hydrophobic residues namely, L83[2.60], L104[3.29] and L107[3.32] are found lining the orthosteric pocket of the receptor, and together, they form the microenvironment for the binding of the ligands. The interactions between niacin, acipimox, MK6892, GSK256073 and MMF with the receptor are mainly ionic, hydrophobic, and aromatic, including residues predominantly from TM2, TM3, TM7 and

ECL2, and a complete list of interactions are listed in Supplementary Fig. 14.

The comparison of the ligand binding pocket in all five structures reveals that R111[3.36] forms the most important residue for binding to the negatively charged carboxyl group of ligands, niacin, acipimox, MK6892, GSK256073 and MMF through hydrogen bond (Figs. 4d, 5a). Previous studies have suggested that this carboxyl group is critical for receptor activation, and substitution with an amide group abolishes GPR109A activity[3], and our structural snapshots provide a mechanistic basis for these functional observations. Three more pairs of hydrogen bonds can be observed, one between the carboxyl moiety of niacin (or acipimox, GSK256073 and MMF) with the side chain Y284[7.43], two between the chloride moiety of GSK256073 with the side chain S179[ECL2] and backbone N-atom of F180[ECL2] (Figs. 4d, 5a) and one between oxo-group at position 4 of MMF with S179[ECL2]. Furthermore, activation of GPR109A appears to require a hydrophobic environment within the orthosteric pocket, and several hydrophobic contacts can be found to stabilize niacin and acipimox within the ligand binding pocket mediated by hydrophobic residues namely, L83[2.60], L104[3.29], L107[3.32], F180[ECL2], F277[3.36], and L280[7.39] (Fig. 5a, b and Supplementary Fig. 14). Similarly, hydrophobic residues such as L104[3.29], L107[3.32], F277[7.36], and L280[7.39] forms extensive interactions with GSK256073 (Fig. 5a, b and Supplementary Fig. 14). Although niacin, acipimox and GSK256073 exhibit hydrophilic, hydrophobic, and charged properties that largely match with those of the ligand binding pocket (Supplementary Fig. 14),

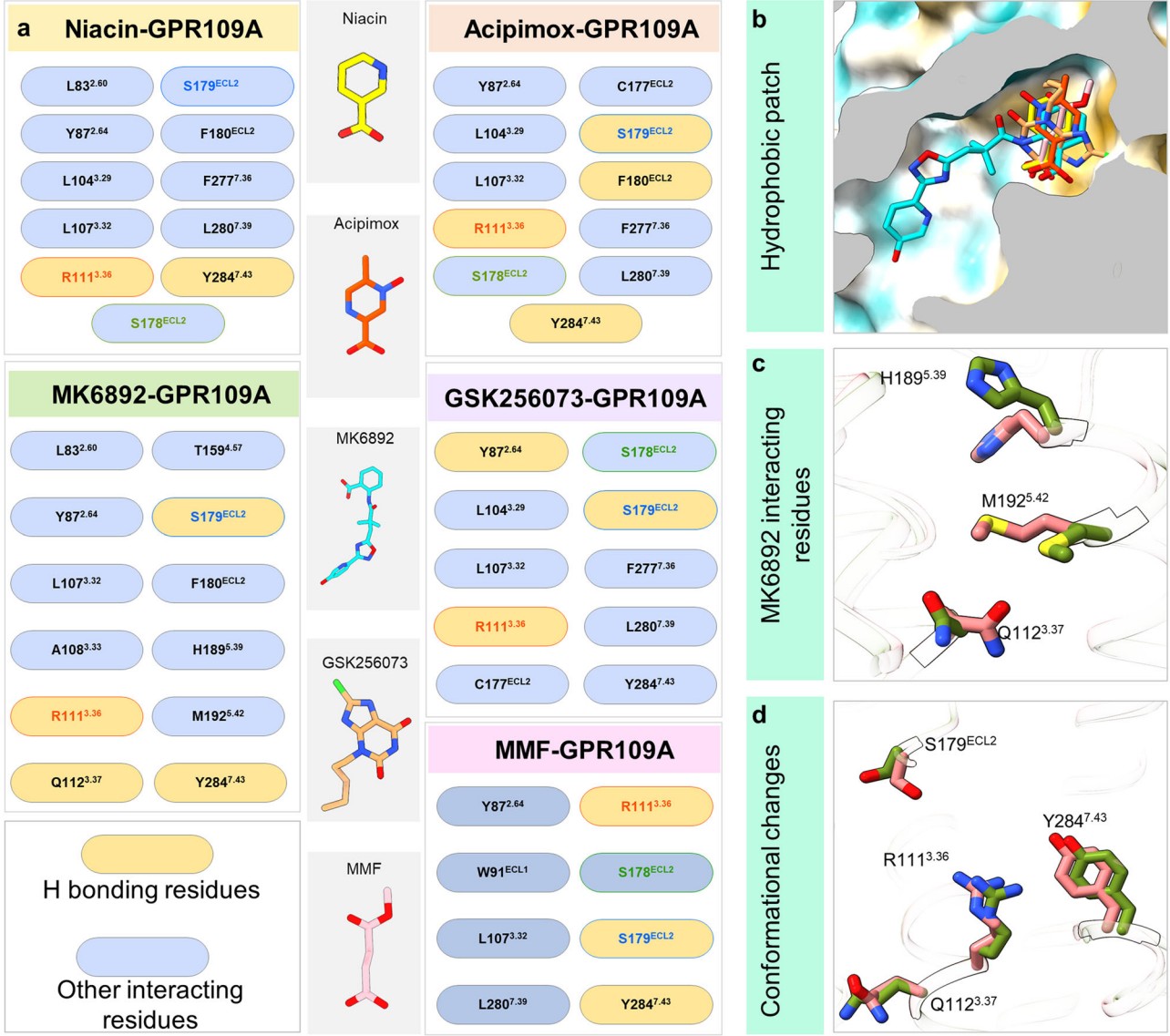

**Fig. 5 | Ligand binding interface of GPR109A. a** List of GPR109A residues interacting with ligands, with key interacting residues highlighted (R111[3.36] – orange, S178[ECL2] – green, S179[ECL2] – blue). **b** Cross-section of GPR109A orthosteric pocket depicting the hydrophobic patch surrounding the individual ligand. **c** Interacting residues of GPR109A with the extended part of MK6892. **d** Conformational changes of GPR109A residues interacting with MK6892 with respect to niacin.

slight conformational variation can be observed within the binding pocket for the niacin or acipimox and GSK256073 complexes (Fig. 4c). These conformational shifts can be attributed to the presence of the extra Cl moiety in GSK256073.

Like niacin, acipimox, MMF and GSK256073, the carboxyl group of MK6892 makes similar contacts with the surrounding polar and hydrophobic residues within the orthosteric binding pocket (Figs. 4d, 5a). MK6892 has a relatively extended chemical structure compared to the other three agonists and therefore, it engages several additional residues in the receptor. For example, the extended moieties in MK6892 i.e., dimethyl, oxadiazole, and pyridyl groups interact with Q112[3.37], H189[5.39] and M192[5.42] in an extended binding pocket in the receptor (Fig. 5c). Interestingly, several conformational rearrangements in the side-chains of R111[3.36], Q112[3.37], S179[ECL2], and Y284[7.43] are also observed compared to the other agonists in order to accommodate the bulky extended group of MK6892 (Fig. 5d). Furthermore, an upward rotameric transition of H189[5.39] and an outward shift of M192[5.42] is also observed within the extended binding pocket to prevent steric clashes with the extended chain of MK6892 (Fig. 5c). These

additional interactions of MK6892 with the GPR109A are similar to those observed in recent studies[29,30].

## Agonist-induced activation of GPR109A

When compared to the recently determined inactive state crystal structure of GPR109A (7ZL9)[29], all the ligand-bound structures displayed conserved conformational changes (Fig. 6a). For example, the cytoplasmic side of TM6 of niacin-activated GPR109A exhibits an outward movement of ~4 Å (measured from the Cα of K227) and 7 Å inward movement of TM5 towards the extracellular side (measured from the Cα of H189[5.39]) and about 3.3 Å outward movement towards the cytoplasmic side (measured from the Cα of R218[ICL3]) (Fig. 6b, c). The agonist-bound structures of GPR109A exhibit the typical hallmark movements of receptor activation as reflected by the conserved motifs and microswitches. For example, the "P-I-F motif" consisting of P200[5.50], I115[3.40] and F240[6.44] forms an interface at the base of the ligand binding pocket, and it undergoes conformational rearrangements upon receptor activation. The rearrangements include: (i) rotameric shift of P200[5.50], (ii) rotameric flip of I115[3.40] and (iii) large transition of F240[6.44], thus opening the cytoplasmic core of the receptor for the

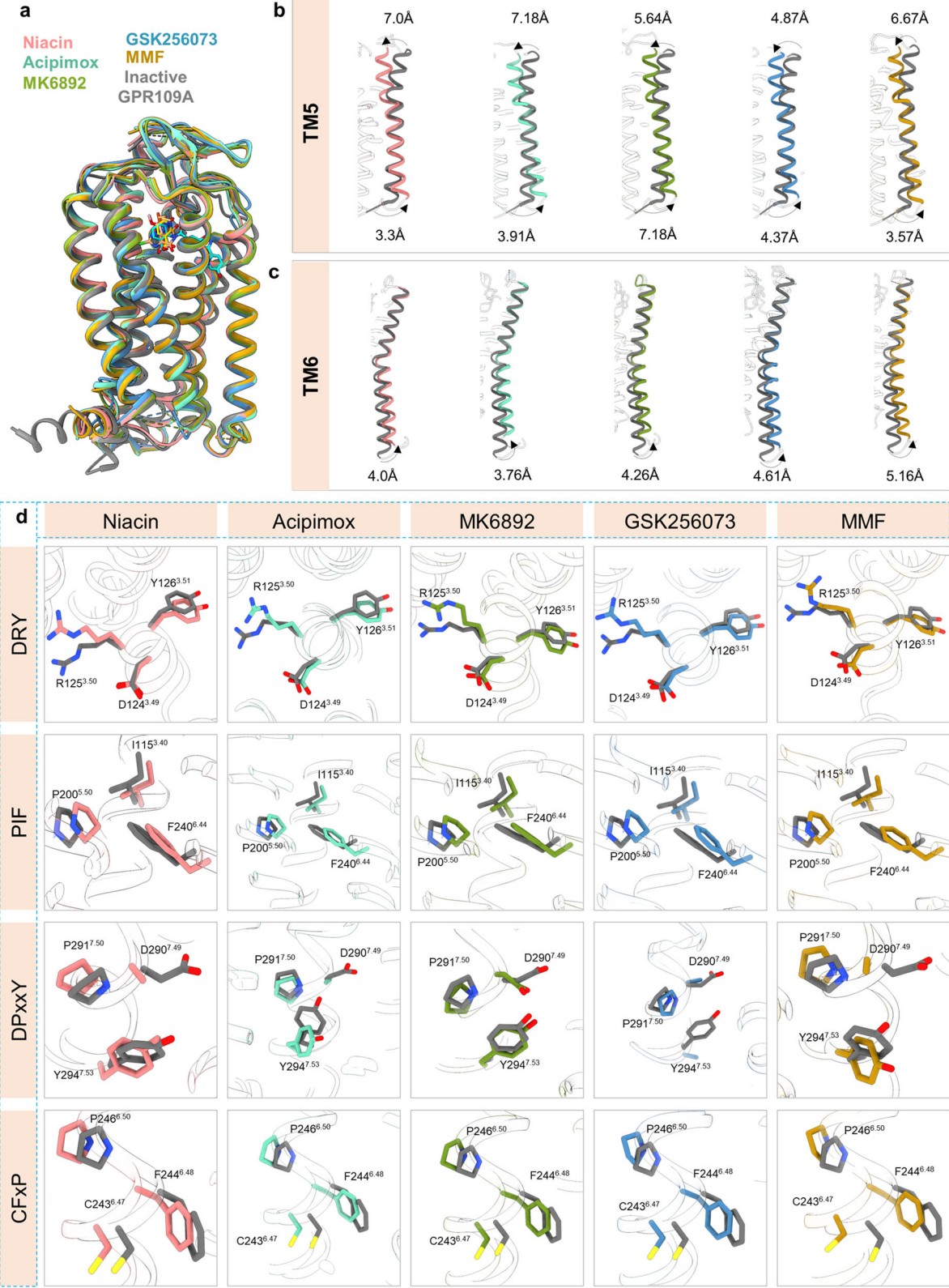

**Fig. 6 | Major conformational changes on GPR109A activation.**
**a** Superimposition of inactive GPR109A with receptor bound to niacin, acipimox, MK6892, GSK256073, and MMF. **b**, **c** Displacements of TM5, TM6 upon GPR109A activation in the structures of niacin, acipimox, MK6892, GSK256073, and MMF bound GPR109A respectively. **d** Conformational changes in the conserved microswitches (DRY, PIF, NPxxY, CW/FxP) in the active structure of GPR109A.

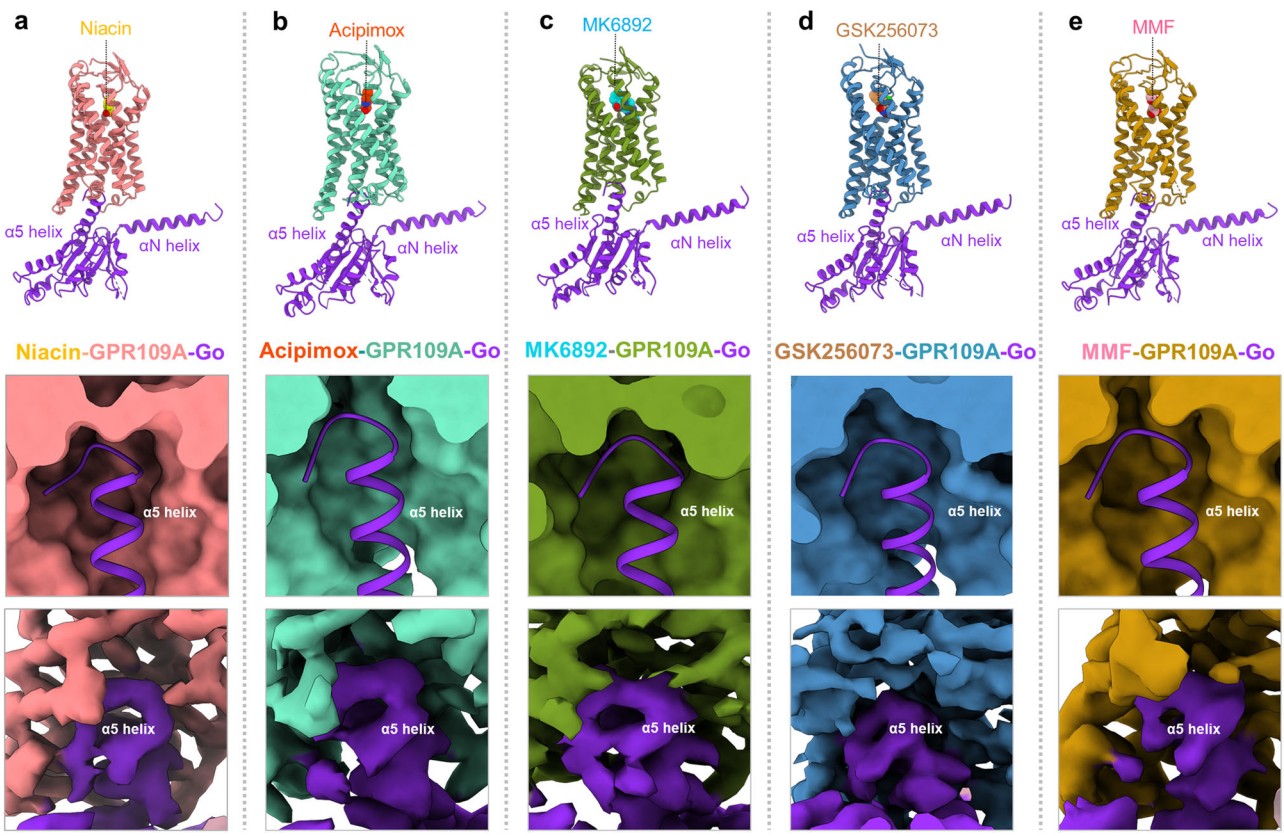

**Fig. 7 | GPR109A-G-protein interacting interface. a–e** Representation of α5 helix of Gαo docking into the cytoplasmic core of GPR109A bound to niacin, acipimox, MK6892, GSK256073 and MMF, respectively.

interaction with the α5 helix residues of Gαo (Fig. 6d). Similar conformational changes can be observed with respect to the "DRY" and "NPXXY" microswitches as well. D124[3.49], R125[3.50] and Y126[3.51] in TM3 is a highly conserved motif where D124[3.49] forms a salt bridge with R125[3.50], thus locking the receptor in an inactive conformation. A rotameric shift of R125[3.50] can be observed in the ligand-bound structures, facilitating the breaking of the salt-bridge/ionic lock and transition to its active conformation (Fig. 6d). A variant of the "NPxxY" motif is present in GPR109A, where N290[7.49] is substituted with D290[7.49] in TM7. Upon activation, the lower portion of TM7 moves inwards towards the receptor core combined with a rotation of Y294[7.53] along the helical axis (Fig. 6d).

### GPR109A-G-protein interaction interface

We observe significant movements of TM5, TM6 and TM7 create an opening on the cytoplasmic surface of the receptor that allows the docking of the α5 helix of Gαo leading to coupling of G-proteins with the activated receptor (Fig. 7a–e). Expectedly, we observe a large buried surface area at the interface of the receptor and G-protein nearing almost 2,000 Å[2] as typically observed in GPCR-G-protein complexes, and this is almost identical in all five structures of GPR109A reported here (Fig. 7a–e). The GPR109A-G-protein interface is stabilized by extensive hydrophobic and polar interactions between the TM2, TM3, ICL2, ICL3, TM6, TM7 and Helix 8 in the receptor and the α5 helix of Gαo (Fig. 8a–j). Specifically, Y354 at the carboxyl-terminus of Gαo forms a key residue that is positioned in pocket on the cytoplasmic side of the receptor lined by K225[6.29], I226[6.30] and P299[8.48] (Fig. 8a). In addition, several hydrogen bonds between D341, N347 and G352 of Gαo with R218[ICL3] and R128[3.53] of the GPR109A, respectively, further stabilize the interaction (Fig. 8a–e). Finally, the stretch from A135 to K138 in the ICL2 of the receptor adopts a one-turn helix where H133[ICL2] interacts with T340 which lie within a hydrophobic pocket

formed by the residues from α5 helix, αN-β1 loop and β2-β3 loop of Gαo (Fig. 8c). The receptor-G-protein interface is further stabilized by residues of ICL3 with α5 C-terminal loop and α4-β6 loop of Gαo, viz. R218[ICL3] forms extensive interactions with T340, D341 and D337 of Gαo (Fig. 8a–e). A list of ligand bound-GPR109A residues interacting with Gαo is presented in Fig. 8f–j, which underscores a largely conserved interface for G-protein interaction although some ligand-specific interactions are also observed. A comprehensive detail of the interactions between GPR109A and G-proteins are listed in Supplementary Fig. 15.

### Structure-guided design of receptor inactivation and biased-agonism

There were two key interactions in the ligand binding pocket namely the R111 in TM3 and S179 in ECL2 that appeared to be conserved in all five structures with various ligands (Fig. 9a). Therefore, we generated R111[3.36]A and S179[ECL2]A mutants of the receptor and measured single-dose activation of G-protein and βarr-coupling vis-à-vis the wild-type receptor in response to niacin, GSK256073, acipimox, MMF, and MK6892. These mutants expressed at comparable levels as the wild-type receptor (Supplementary Fig. 16). We observed that the R111[3.36]A mutant showed complete loss of both G-protein activation and βarr recruitment in response to niacin, acipimox, GSK256073, and MMF, whereas MK6892 showed G-protein activation but remained silent on βarr recruitment (Fig. 9b, c). On the other hand, the S179[ECL2]A mutant maintained cAMP response and βarr recruitment robustly (Fig. 9b, c). Next, we carried out the dose-response experiments on the R111[3.36]A and S179[ECL2]A mutants in response to various agonists in order to better understand their activation and transducer-coupling. We observed that stimulation of R111[3.36]A with niacin fails to result in measurable cAMP response and βarr recruitment (Fig. 9d, e), which is in agreement with a

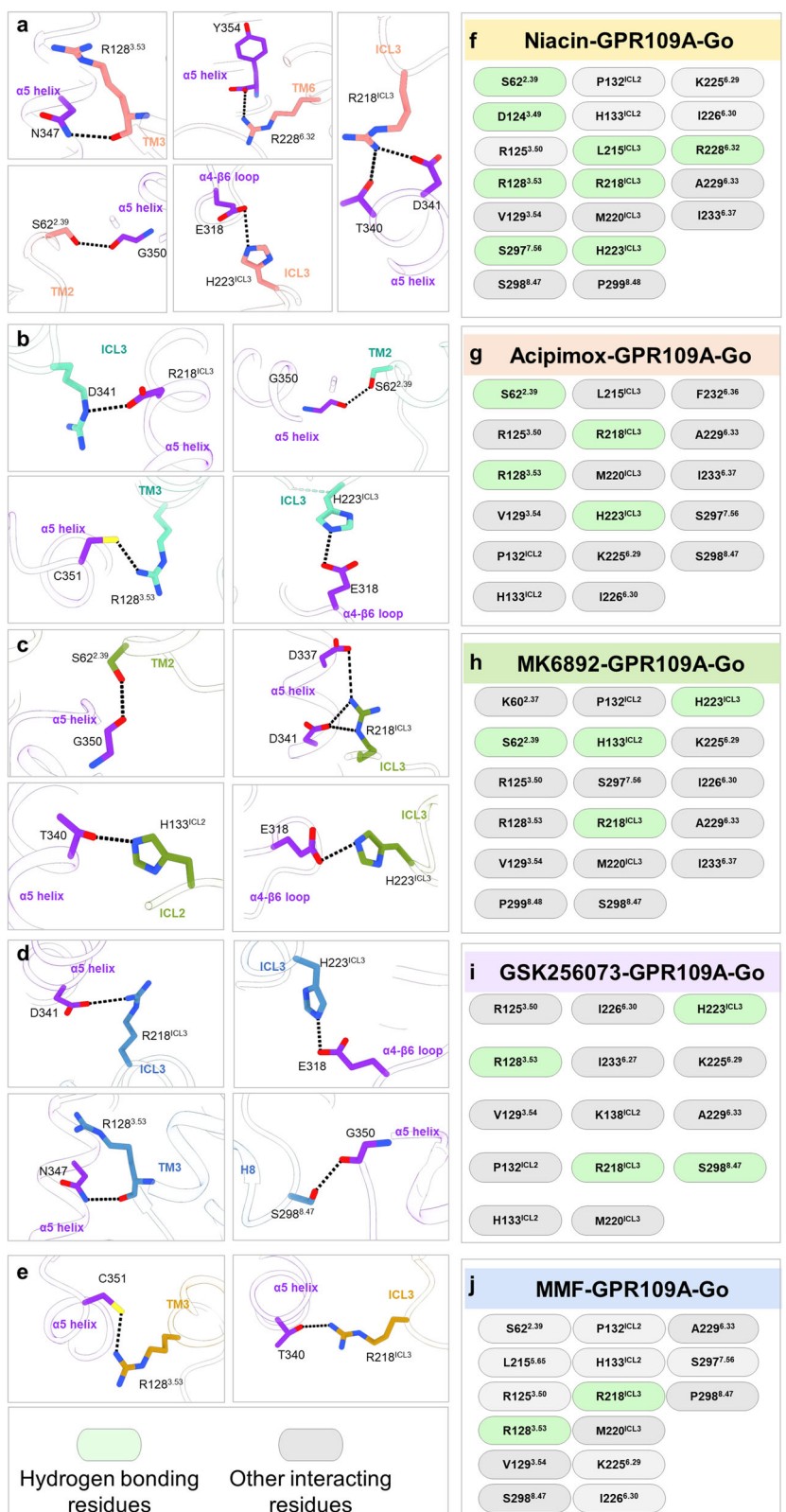

**Fig. 8 | GPR109A-G-protein interacting residues. a–e** Key interactions between Gαo residues and residues of the cytoplasmic core of GPR109A. Black dotted line represents the H-bond. **f–j** List of GPR109A residues interacting with Gαo in niacin, acipimox, MK6892, GSK256073, and MMF bound structures.

previous study suggesting near-complete loss of niacin binding to this mutant[2]. On the other hand, the S179$^{ECL2}$A mutant exhibits improved G-protein-coupling (E$_{max}$ 25.78 vs. 64.95, EC$_{50}$ 259 ± 125 nM vs. 9.90 ± 1.42 nM for WT vs. S179$^{ECL2}$A) but a reduction in βarr recruitment (E$_{max}$ 1.61 vs. 1.47, EC$_{50}$ 32.0 ± 4.90 nM vs.

305 ± 65 nM for WT vs. S179$^{ECL2}$A) (Fig. 9d, e). Therefore, GPR109A$^{S179A}$ represents a G-protein-biased mutant for niacin, and it may be a useful tool to probe the intricate details of signaling-bias at this receptor in future studies. We also observed that unlike other agonists, MK6892 is able to activate G-protein-coupling to R111$^{3.36}$A

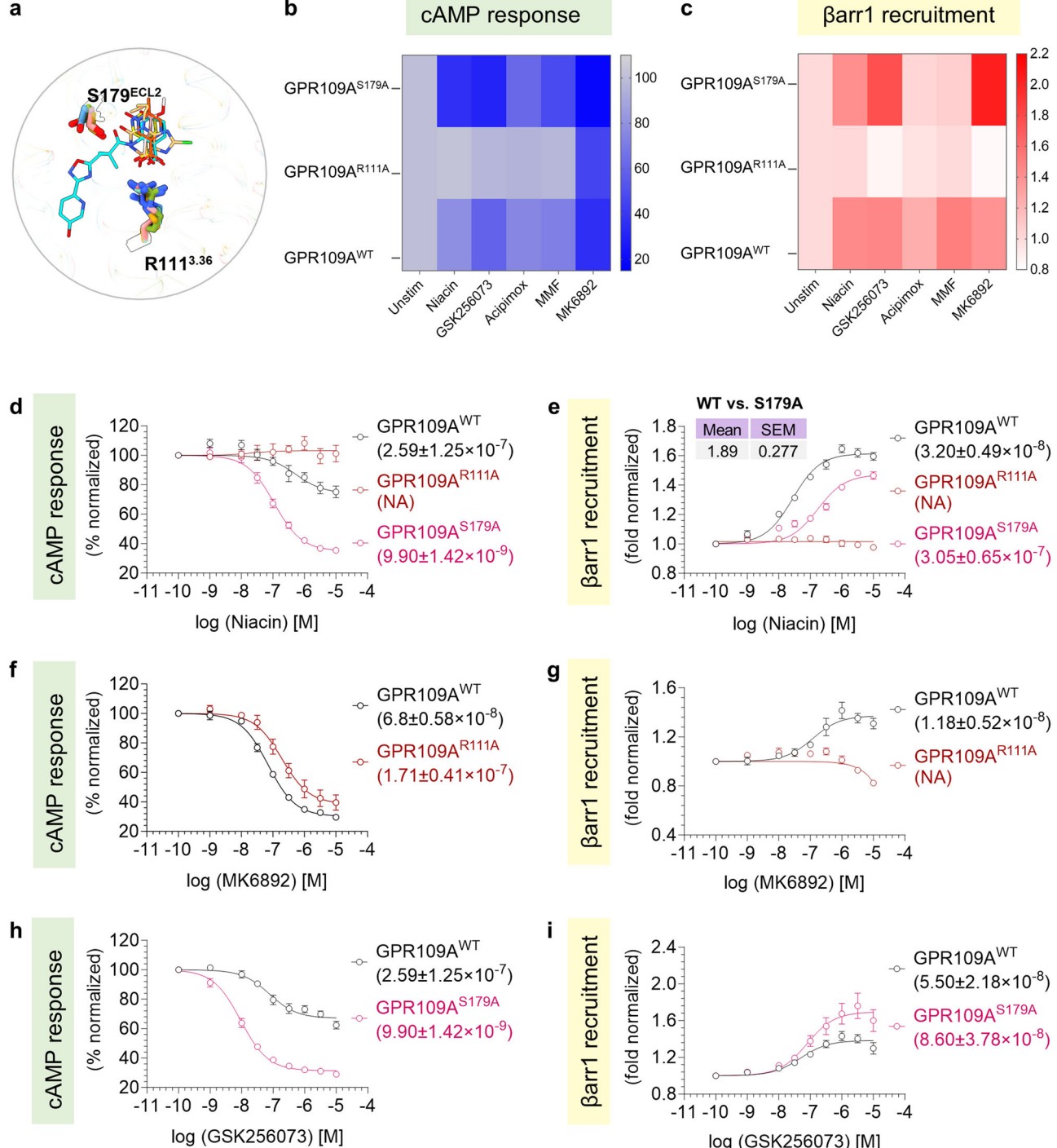

**Fig. 9 | Contribution of orthosteric pocket residues on GPR109A mediated signaling. a** Cartoon representation of residues interacting via H-bond with niacin (yellow), acipimox (orange red), MK6892 (cyan), GSK256073 (sandy brown), and MMF (light pink). **b-c** cAMP response downstream of GPR109A[WT], GPR109A[R111A], and GPR109A[S179A] in response to the indicated ligands was studied by GloSensor assay (**b**) (mean ± SEM; $n = 3$ independent experiments; % normalized with the minimum concentration for each ligand as 100) and NanoBiT-based βarr1 recruitment assay (**c**) (mean ± SEM; $n = 3$ independent experiments; fold normalized with the minimum concentration for each ligand as 1). **d, f, h** cAMP decrease studied by Glo-Sensor assay downstream of GPR109A[WT], and mutants in response to niacin, MK6892, and GSK256073 (mean ± SEM; $n = 3$–5 independent experiments; $n = 3$ in response to niacin and MK6892 and $n = 5$ in response to GSK256073; % normalized with the minimum concentration for each ligand as 100). **e, g, i** NanoBiT-based

βarr1 recruitment downstream of GPR109A[WT], and mutants in response to niacin, MK6892, and GSK256073 (mean ± SEM; $n = 3$–4 independent experiments; $n = 3$ in response to MK6892, $n = 4$ in response to niacin, and GSK256073; fold normalized with the minimum concentration for each ligand as 1). Bias factor for the mutant GPR109A[S179A] (shown in the inset of **e**) was calculated using the software https://biasedcalculator.shinyapps.io/calc/ (Detailed formula is provided in methods section). The compiled data of G-protein activation and βarr1 recruitment assays (mean ± SEM, $n = 3$–4 independent experiments; $n = 3$ for cAMP response and $n = 4$ for βarr1 recruitment) was used for the calculation. During bias factor calculation GPR109A[WT] was considered as reference and observed G-protein bias with GPR109A[S179A] upon stimulation with niacin. Source data is provided as Source Data file.

mutant as measured using cAMP response although it fails to elicit measurable βarr recruitment (Fig. 9f, g). On the other hand, stimulation of S179^ECL2A mutant with GSK256073 results in a robust enhancement in cAMP response as well as a modest augmentation of βarr recruitment compared to the wild-type receptor (Fig. 9h, i).

## Discussion

While this manuscript was under review, additional studies reporting GPR109A structures were published[29–36], and therefore, we compare the structures presented in the current study with those published recently. We observed that the overall features of ligand binding, receptor activation, and G-protein-coupling are mostly similar (Supplementary Fig. 17). Taken together, these contemporary studies provide mutually aligned conclusions and insights into GPR109A activation and signaling that should facilitate further research on this therapeutically important receptor.

While niacin, acipimox and MMF bind in a similar pose in the orthosteric binding pocket and make nearly identical interactions, MK6892 and GSK256073 make additional contacts in the ligand binding pocket as expected based on their extended chemical structures. Interestingly, both MK6892 and GSK256073 appear to exhibit enhanced G-protein-coupling compared to niacin. However, GSK256073 is relatively more efficacious and potent in βarr recruitment assay while MK6892 is weaker than niacin in the βarr recruitment assay. Based on these data, it is plausible that additional contacts made by MK6892 impart a greater potency and efficacy compared to niacin, although follow-up experiments are required to test this hypothesis. The G-protein-bias of MK6892 may explain its non-flushing properties although transducer-coupling profile of GSK256073 does not align with this same hypothesis, and therefore, segregation of lipolysis vs. flushing response through GPR109A may involve additional level of fine-tuning that remains to be determined.

Guided by structural visualization of the key interactions between the agonists and the receptor, we engineer two mutants namely, R111^3.36A and S179^ECL2A. The mutant R111^3.36A failed to induce measurable transducer-coupling upon stimulation by niacin, acipimox, MMF and GSK256073 suggesting a key contribution of this residue in agonist-binding, receptor activation and signaling (Fig. 10). On the other hand, S179^ECL2A mutant displays an enhanced G-protein-coupling upon niacin-stimulation as measured using cAMP response but impaired βarr-recruitment, making it G-protein-biased (Fig. 10). It is also worth noting that in contrast with niacin, MK6892 is still able to elicit G-protein activation through R111^3.36A mutant while GSK256073 results in augmented transducer-coupling for both, G-protein and βarr

compared to the wild-type receptor. These observations suggest that agonists with distinct chemical structures can modulate activation and transducer-coupling responses at GPR109A differently. Moreover, as S179 is located in the ECL2, away from the direct transducer-coupling interface, these data also suggest that S179^ECL2A mutation modulates G-protein- and βarr-coupling through allosteric mechanisms as reported earlier for other GPCRs[37,38]. Taken together, these data demonstrate the feasibility of structure-guided engineering of receptor inactivation and biased-agonism at the level of transducer-coupling responses.

In summary, the structural snapshots of GPR109A presented here elucidate the molecular details of the interaction of chemically diverse agonists, uncover the mechanism of activation and signaling-bias, and guide the design of a G-protein-biased mutant of the receptor. Our findings could potentially pave the way for rational therapeutic design targeting GPR109A, and they also provide a framework to impart signaling bias in GPCRs guided by structural insights that may help deconvolute the mechanism of biased agonism going forward.

## Methods

### General reagents, plasmids, and cell culture

The majority of standard reagents were purchased from Sigma Aldrich unless mentioned. Dulbecco's Modified Eagle's Medium (DMEM), Phosphate Buffer Saline (PBS), Trypsin-EDTA, Fetal-Bovine Serum (FBS), Hank's Balanced Salt Solution (HBSS), and Penicillin-Streptomycin solution were purchased from Thermo Fisher Scientific. HEK-293 cells were purchased from ATCC and maintained in 10% (v/v) FBS (Gibco, Cat. no. 10270-106) and 100 μg ml^-1 penicillin and 100 μg ml^-1 streptomycin (Gibco, Cat. no. 15140122) supplemented DMEM (Gibco, Cat. no. 12800-017) at 37 °C under 5% CO$_2$. The cDNA coding region of GPR109A^WT, GPR109A^R111A, and GPR109A^S179A, with a HA signal sequence, a FLAG tag followed by the N-terminal region of M4 receptor (2–23 residues) at the N-terminus was cloned into pcDNA3.1 vector. For GloSensor assay, luciferase-based 22 F cAMP biosensor construct was purchased from Promega. For the constructs used in NanoBiT assay, SmBiT was fused at the C-terminus of the receptor, and the LgBiT- βarr1/2 construct was the same as previously described[39]. GPR109A orthosteric binding pocket mutants were generated by site-directed mutagenesis using Q5 Site-Directed Mutagenesis Kit (NEB, Cat. no. E0554S). All DNA constructs were verified by sequencing from Macrogen. The oligonucleotides used in this study to generate constructs are listed in Supplementary Table 1. Niacin was purchased from HiMedia (Cat. no. TC157), acipimox and MMF were purchased from Sigma Aldrich (Cat. no. 92571 and Cat. no. 651419), respectively. GSK256073 and MK6892 were purchased from MedChemExpress (Cat. no. HY10680 and HY119222, respectively).

### GPR109A purification

Codon-optimized human GPR109A gene was cloned into the pVL1393 vector with an N-terminal HA signal sequence, a FLAG tag, and a 21-amino-acid stretch of the M4 receptor N-terminal region (amino acids 2–23, ANFTPVNGSSGNQSVRLVTSSS) for increased expression. The receptor was expressed and purified from *Spodoptera frugiperda* (Sf9) cells using a baculovirus-mediated expression system. Briefly, Sf9 cells were co-transfected with GPR109A-pVL1393 construct and BestBac baculovirus DNA (Expression Systems) to produce recombinant baculovirus, which was subsequently amplified and used for GPR109A expression. For receptor purification, insect cells were infected with recombinant baculovirus for 72 h at 27 °C and harvested by high-speed centrifugation. Post-harvest, insect cells were sequentially dounced in hypotonic buffer (20 mM HEPES, pH 7.4, 20 mM KCl, 10 mM MgCl$_2$, 1 mM PMSF, and 2 mM Benzamidine), hypertonic buffer (20 mM HEPES, pH 7.4, 1 M NaCl, 20 mM KCl, 10 mM MgCl$_2$, 1 mM PMSF, and 2 mM Benzamidine) and lysis buffer (20 mM HEPES, pH 7.4, 450 mM NaCl, 1 mM PMSF, and 2 mM Benzamidine). Lysed cells were solubilized by continuous

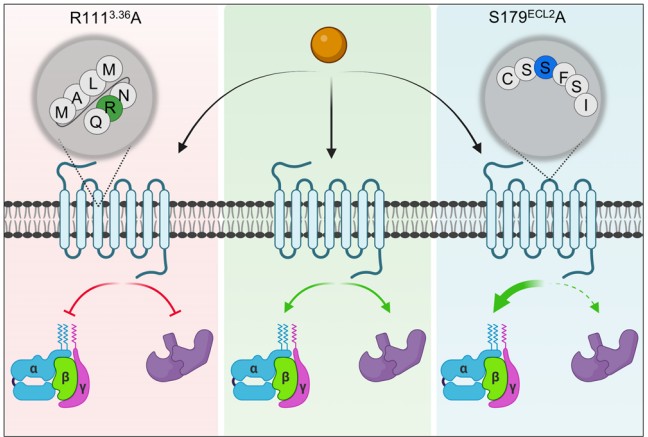

**Fig. 10 | Structure guided bias signaling.** Schematic depicting the effect of two mutants GPR109A^S179A and GPR109A^R111A in G-protein activation and βarr1 recruitment in response to niacin. Source data is provided as source data file. (Created with BioRender.com).

stirring in 1% L-MNG (Anatrace, Cat. no. NG310) for two hours at 4 °C in the presence of 0.01% cholesteryl hemisuccinate (Sigma, Cat. no. C6512) and 2 mM Iodoacetamide. After solubilization, salt concentration was lowered to 150 mM and cell debris was separated by high-speed centrifugation at $47,850 \times g$, 4 °C and the receptor was enriched on M1-anti FLAG columns. Non-specific proteins were removed by three washes of low salt buffer (20 mM HEPES, pH 7.4, 150 mM NaCl, 0.01% MNG, 0.01% CHS, and 2 mM $CaCl_2$) alternating with two washes of high salt buffer (20 mM HEPES, pH 7.4, 350 mM NaCl, 0.01% MNG, and 2 mM $CaCl_2$). The bead-bound receptor was eluted with FLAG peptide-containing buffer (20 mM HEPES, pH 7.4, 150 mM NaCl, 2 mM EDTA, and 250 μg ml$^{-1}$ flag peptide). The purified receptor was incubated with 2 mM iodoacetamide for 30 min on ice. Afterwards, 2 mM L-cysteine was added and incubated it for an additional 30 min. The purified receptor was concentrated using a 30 kDa MWCO concentrator (Cytiva, Cat. no. 28932361) and stored at −80 °C with a 10% final glycerol concentration. 1 μM of each ligand was kept throughout the purification.

### Purification of Gβ1γ2 dimer
N terminal 6X His tagged Gβ1 subunit and untagged Gγ2 subunits were cloned in the Dual pVL1392 vector and expressed in the Sf9 cell using a baculovirus-based expression system as explained for receptor. After harvesting, cells were lysed by douncing in lysis buffer (50 mM Tris-Cl, pH 8.0, 150 mM NaCl, 1 mM PMSF, and 2 mM Benzamidine) and pelleted by centrifugation at $47,850 \times g$ for 20 min at 4 °C. The cell pellet was dounced in solubilization buffer (50 mM Tris-Cl, pH 8.0, 150 mM NaCl, 5 mM β-mercaptoethanol, 1% DDM (Anatrace, Cat. no. D310), 1 mM PMSF, and 2 mM Benzamidine) and solubilized for 2 hrs at 4 °C with constant stirring. Cell debris was separated by centrifugation at $47,850 \times g$ and cell lysate was passed through the Ni-NTA column using gravity flow. Non-specifically bound proteins were removed by a one-two column wash with buffer (50 mM Tris-Cl, pH 8.0, 150 mM NaCl, 0.01% MNG), and protein was eluted with 250 mM Imidazole (50 mM Tris-Cl, pH 8.0, 150 mM NaCl, 0.01% MNG, 250 mM Imidazole). Eluted protein was concentrated with a 10 kDa MWCO concentrator (Cytiva Cat. no. 28932360) and stored with 10% final glycerol concentration.

### Mini Gαo purification
The gene encoding miniGαo was designed as described previously[40,41] and cloned into pET-15b (+) vector with 6X His-tag at the N-terminal followed by TEV protease cleavage site. The recombinant construct was transformed into *E. coli* BL21(DE3) cells. A 5 ml starter culture, grown for 6–8 h at 37 °C, was inoculated into a 50 ml primary culture media supplemented with 0.2% glucose and allowed to grow at 30 °C for 16–18 h. 1.5 litre of TB (Terrific Broth) media was inoculated with 15 ml of primary culture and grown at 30 °C. At O.D$_{600}$ 0.8, cells were induced with 50 μM IPTG (Isopropyl β-D-1-thiogalactopyranoside) and allowed to grow for an additional 18–20 h. Harvested cells were lysed by lysozyme in lysis buffer (40 mM HEPES, pH 7.4, 100 mM NaCl, 10 mM Imidazole, 10% Glycerol, 5 mM $MgCl_2$, 1 mM PMSF, 2 mM Benzamidine, 50 μM GDP, 100 μM DTT, and 1 mg ml$^{-1}$ lysozyme), followed by disruption by ultrasonication. Cell debris was pelleted by high-speed centrifugation at 4 °C, and protein was enriched on the Ni-NTA column. Non-specifically bound proteins were removed by extensive washing (20 mM HEPES, pH 7.4, 500 mM NaCl, 40 mM Imidazole, 10% Glycerol, 50 μM GDP, and 1 mM $MgCl_2$), and protein was eluted with 500 mM Imidazole (in 20 mM HEPES, pH 7.4, 100 mM NaCl, and 10% Glycerol). His-tag was cleaved by overnight TEV treatment at room temperature (1:20, TEV: protein), and untagged protein was recovered by size exclusion chromatography on Hi-Load Superdex 200 PG 16/600 column (Cytiva, Cat. no. 17517501). Fractions corresponding to cleaved protein were pooled, analyzed on SDS-PAGE, and stored at −80 °C with 10% glycerol.

### ScFv16 purification
The gene encoding ScFv16 was cloned in pET-42a (+) vector downstream of 10X His-tagged MBP gene with a TEV protease cleavage site between them and overexpressed in the *E. coli* Rosetta (DE3) strain[42]. A single colony from a freshly transformed plate was inoculated in 50 ml of 2XYT media and allowed to grow overnight at 37 °C. 1 L 2XYT media supplemented with 0.5% glucose and 5 mM $MgSO_4$ was inoculated with overnight primary culture and induced with 250 μM IPTG at O.D$_{600}$ of 0.8–1.0 and allowed to grow for 16–18 h at 18 °C. Post-harvest, cells were resuspended in 20 mM HEPES, pH 7.4, 200 mM NaCl, 30 mM Imidazole, 1 mM PMSF, and 2 mM Benzamidine buffer and incubated at 4 °C for 40 min with constant stirring. Cells were lysed by sonication, and cell debris was removed with high-speed centrifugation at 4 °C. Protein was captured on the Ni-NTA column using gravity flow, and non-specific proteins were removed by extensive washing with 20 mM HEPES, pH 7.4, 200 mM NaCl, and 50 mM Imidazole. Bound ScFv16 was eluted with 300 mM imidazole-containing buffer (20 mM HEPES, pH 7.4, 200 mM NaCl) and was re-passed through amylose resins, and after one column wash with 20 mM HEPES, pH 7.4, 200 mM NaCl buffer, bound protein was eluted with 10 mM maltose (prepared in 20 mM HEPES, pH 7.4, 200 mM NaCl). To obtain tag-free ScFv16, the eluted protein was overnight digested with TEV protease, and His-MBP was removed by passing the digested protein through the Ni-NTA column. Eluted protein was further cleaned by size exclusion chromatography on Hi-Load Superdex 200 PG 16/600 column, analysed on SDS-PAGE and stored at −80 °C with 10% glycerol.

### Reconstitution of GPR109A-G-protein-ScFv16 complexes
Purified GPR109A was mixed with a 1.2 molar excess of miniGo, Gβγ, and ScFv16 in the presence of 25 mμ ml$^{-1}$ apyrase (NEB, Cat. no. M0398S) and 1 μM of individual ligand, and complexing was allowed for two hours at room temperature. The receptor complex was concentrated with a 100 kDa MWCO (Cytiva, Cat. no. GE28-9323-19) concentrator and separated from the unbound component by size exclusion chromatography on Superose 6 increase 10/300 GL column (Cytiva, Cat. no. 29091596). The SEC eluate was analyzed on 12% SDS-PAGE, and complex fractions were concentrated to ~10 mg ml$^{-1}$ and stored at −80 °C.

### Negative stain electron microscopy
Homogeneity of the purified protein complexes was determined through negative staining with uranyl formate prior to data collection under cryogenic conditions following the protocols described previously[43,44]. 3.5 μl of the purified complexes were dispensed onto fresh glow discharged carbon/formvar coated 300 mesh Cu grids (PELCO, *Ted Pella*) at a concentration of 0.02 mg ml$^{-1}$ and incubated for 1 min at room temperature. This was followed by blotting off the excess samples from the grids using filter paper. The grid containing the adhered sample was touched onto a first drop of freshly prepared 0.75% uranyl formate stain and immediately blotted off by touching the edge of the grid onto a filter paper. The grid was then touched and incubated on a second drop of uranyl formate for 30 s and left on the bench in a petri plate for air drying. Data collection was performed on a FEI Tecnai G2 12 Twin TEM (LaB6) operating at 120 kV and equipped with a Gatan CCD camera (4k × 4k) at 30,000x magnification. Data processing of the collected micrographs was performed with Relion[45] 3.1.2 version. More than 10,000 particles were autopicked with the gaussian blob picker, extracted and subjected to reference-free 2D classification.

### Cryo-EM sample preparation and data collection
3 μl of the individual complexes at final concentrations in the range of 1.5–3 mg/ml were applied onto glow-discharged Quantifoil holey carbon grids (Cu R2/1 or R2/2) and vitrified in liquid ethane (−181 °C) using a Leica GP plunger (Leica Microsystems, Austria) maintained at 90%

humidity and 10 °C with a blotting time of 3 s. CryoEM movies were acquired on a TFS Glacios microscope operating at 200 kV and equipped with Gatan K3 direct electron detector (Gatan Inc.). Images were collected automatically with SerialEM software in counting mode at a nominal magnification of 46,000x and pixel size of 0.878 Å over a defocus range of 0.5–2.5 μm. An accumulated dose of 55 e$^-$/A$^2$ was fractionated into a movie stack consisting of 40 frames.

## Cryo-EM data processing

All data processing steps were performed with cryoSPARC[46] v4.0 unless otherwise stated. Dose fractionated movie stacks were subjected to beam-induced motion correction using Patch motion correction (multi) followed by estimation of contrast transfer function parameters with Patch CTF estimation (multi). For particle picking, the blob-picker (template free) sub-program was used to automatically pick particles with the circular blob in the diameter range of 100–220 Å.

For the Niacin-GPR109A-Go dataset, 11,070 dose weighted, motion-corrected micrographs were selected for downstream processing. The blob-picker sub-program was used to automatically pic particles with the circular blob in the diameter range of 100–200 Å. This yielded 7,027,107 particles which were subjected to several rounds of reference-free 2D classification to eliminate particles with poor features. 1,737,725 particle projections corresponding to the 2D averages with clear secondary features were selected and subjected to ab-initio reconstruction with 3 classes. Subsequent heterogeneous refinement yielded a model with features of a typical GPCR-G-protein complex containing 1,011,301 particle projections which accounted for 75% of the particles used for heterogeneous refinement. This particle stack was subjected to non-uniform refinement, followed by local refinement with mask excluding the noise outside the molecule, yielding a density map with an indicated global resolution of 3.37 Å at 0.143 FSC cut-off.

For the Acipimox-GPR109A-Go dataset, 9,115,816 particles were autopicked from 11,263 micrographs, extracted with a box size of 360 px (fourier cropped to 64 px) followed by 2D classification to obtain classes with clear secondary features. 1,733,846 particles corresponding to the clean classes were re-extracted with a box size of 360 px (fourier cropped to 288) and subjected to ab-initio reconstruction and heterogenous refinement to generate 4 classes. 1,059,994 particles from the best 3D class were selected and subjected to non-uniform refinement to yield a final reconstruction at a resolution of 3.45 Å.

For the MK6892-GPR109A-Go dataset, 9,068,321 particles were autopicked from 10,753 motion corrected micrographs which were extracted with a box size of 360 px (fourier cropped to 64 px) and subjected to multiple rounds of reference-free 2D classification. 2D class averages consisting of 1,829,840 particles with clear secondary features and resembling conformation of protein complexes were re-extracted with a box size of 360 px and fourier cropped to 288 px. These particle stacks from the extraction job were subsequently subjected to ab-initio reconstruction, followed by heterogeneous refinement yielding 4 models. 1,200,513 particle projections (accounting for 66% of the total particles) from the best 3D class were selected and subjected to non-uniform refinement, which yielded a map with an overall resolution of 3.3 Å using the 0.143 FSC criterion.

For the GSK256073-GPR109A-Go dataset, 5,761,414 particles were automatically picked from 10,574 motion-corrected micrographs. These particle projections were extracted with a box size of 360 px (fourier cropped to 64 px) and subjected to iterative rounds of reference-free 2D classification to discard noisy particles. 1,601,694 particles corresponding to the 2D classes with evident features of protein complexes were selected and re-extracted with a box size of 360 px and fourier cropped to 288 px. These selected particle projections were used to generate 3 maps for heterogeneous refinement. One of the 3D classes with 523,816 particles showing all the features of a GPCR-G-protein complex was subjected to 3D non-uniform refinement, reaching a nominal resolution of 3.45 Å.

For the MMF-GPR109A-Go dataset, autopicking was performed with the blob-picker subprogram which yielded 9,029,435 particles. Particles were extracted with a box size of 360 px (fourier cropped to 64 px) and pared down to 1,058,559 particles after reference-free 2D classification. The clean particle stack was then re-extracted with a box size of 360 px (fourier cropped to 288 px). Two rounds of ab initio and subsequent hetero-refinement (using four models) were then performed to further refine the particle stack to 678,286 particles. Non-uniform refinement and successive local refinement resulted in a map with an estimated global resolution of 3.75 Å at 0.143 cut-off.

Local resolution estimation of all maps was determined using the Blocres subprogram within cryoSPARC with the corresponding half maps. Sharpening of all maps was performed with "Autosharpen maps" within the Phenix suite[47,48] to enhance features for model building.

## Model building and refinement

Coordinates from an AlphaFold model of GPR109A (AF-Q8TDS4-F1) was used to dock into the EM density map of niacin-GPR109A-Go using Chimera[49,50]. Similarly, coordinates of Gαo, Gβγ and ScFv16 were obtained from a previously solved structure of C5aR1 in complex with Gαo (PDB: 8HPT). The combined model so obtained containing all the components was subjected to "all atom refine" sub-module within COOT[51], followed by manual rebuilding of the residues and the ligand. The rebuilt model was subjected to real space refinement in Phenix[47,48] to obtain a model with 96.94% of the residues in the most favored region and 3.06% in the allowed region of the Ramachandran plot. Validation of all the models was performed with Molprobity[52] within Phenix.

The ligand free model of niacin-GPR109A-Go complex (PDB: 8IY9) was fitted into the density maps of acipimox-GPR109A-Go and MMF-GPR109A-Go in Chimera followed by flexible fitting of the coordinates with the "all atom refine" module in COOT. After several rounds of manual adjustments, the generated model was automatically refined with Phenix_refine. The final refined models of acipimox-GPR109A-Go and MMF-GPR109A-Go showed good Ramachandran statistics with 96.90% and 96.63% in the most favored regions of the Ramachandran plot, respectively.

Likewise, the ligand-free model of niacin-GPR109A-Go complex (PDB: 8IY9) was used to dock into the density maps of MK6892-GPR109A-Go and GSK256073-GPR109A-Go using Chimera. The docked model and the corresponding maps were then imported into COOT and fitted into the respective maps with the "all atom refine" module. The poorly fitted regions were manually adjusted in COOT followed by iterative refinement of the coordinates against the maps using Phenix_refine. The final refined models of MK6892-GPR109A-Go and GSK256073-GPR109A-Go contained residues in 97.73% and 96.73% of the most favored regions of the Ramachandran plot with no outliers. Data collection, processing and model refinement statistics are provided in Supplementary Fig. 10. All figures included in the manuscript have been prepared with Chimera and ChimeraX software.

## Transient transfection of HEK-293 cells

For the GloSensor and NanoBiT assays described in the following subsections, HEK-293 cells were seeded at a density of 3 million in 100 mm cell culture plate (Corning, Cat. no. 3296) 24 h prior to transfection. Transfection mix was prepared by combining the plasmid DNA and polyethyleneimine linear (PEI) (Polysciences, Cat. no. 23966) in 500 μl serum-free DMEM and incubated for 10 min in a microcentrifuge tube. During the incubation of the transfection mixture, FBS supplemented media of the culture plate was replaced with serum-free DMEM, followed by addition of the transfection mixture to the cells.

## GloSensor-based cAMP assay

cAMP response upon ligand stimulation was measured by GloSensor assay[53]. Briefly, HEK-293 cells were transfected using 2 µg of GPR109A construct together with 5 µg 22 F cAMP plasmid using the PEI linear at DNA: PEI ratio of 1:3. After 16–18 h of transfection, cells were harvested followed by resuspension of the cell pellet in assay buffer composed of 1X HBSS, 20 mM of 4-(2-hydroxyethyl)-1-piperazineethanesulfonic acid (HEPES) pH 7.4 and D-luciferin (0.5 mg ml$^{-1}$) (GoldBio, Cat. no. LUCNA-1G). Harvested cells were then seeded in an opaque flat bottom white 96 well cell culture plate (SPL life sciences, Cat. no. 30196) at a density of $2 \times 10^5$ cells well$^{-1}$. After seeding, cells were incubated at 37 °C for 90 min and 30 min at room temperature. After 120 min of incubation basal level luminescence was recorded using a multi-mode plate reader (Lumistar/Fluostar microplate reader, BMG Labtech). To record ligand-induced cAMP decrease as a readout of Gi activation, cellular cAMP level was increased by adding 5 µM forskolin, and luminescence was recorded until the signal got saturated. After that cells were stimulated with corresponding ligands, and luminescence values were recorded for 20 cycles. For stimulation, ligand concentrations ranging from 100 pM to 10 µM were prepared by serial dilution in the buffer constituted of 1X HBSS, 20 mM HEPES pH 7.4. Niacin, acipimox, MK6892, GSK256073 and MMF of different concentrations were added to the corresponding wells. Baseline corrected data were normalized with respect to the luminescence signal of minimal concentration of each ligand as 100% and plotted using nonlinear regression analysis in GraphPad Prism v 9.5.0 software.

## Surface expression assay

Plasma membrane expression of receptors in respective assays was measured by whole cell-based surface ELISA as previously discussed[54]. Briefly, transfected cells were seeded at a density of $2 \times 10^5$ cells well$^{-1}$ in 0.01% poly-D-Lysine pre-treated 24-well plate and incubated for 24 h at 37 °C. Post incubation, growth media was aspirated, and cells were washed with ice-cold 1X TBS for once, followed by fixation with 4% PFA (w/v in 1X TBS) on ice for 20 min. Post fixation, cells were washed three times with 1X TBS (400 µl in each wash) followed by blocking with 1% BSA (w/v in 1X TBS) at room temperature for 90 min. After blocking, 200 µl anti-FLAG M2-HRP was added and incubated for 90 min (prepared in 1% BSA, 1:10,000) (Sigma, Cat. no. A8592). Post antibody incubation, to remove unbound antibodies, cells were washed with 1% BSA (prepared in 1X TBS) three times, followed by the development of signal by treating cells with 200 µl TMB-ELISA (Thermo Scientific, Cat no. 34028) until the light blue color appeared. Signal was quenched by transferring the light blue colored solution to a 96-well plate containing 100 µl 1 M H$_2$SO$_4$. The absorbance of the signal was measured at 450 nm using a multi-mode plate reader. Next, cells were incubated with 0.2% Janus Green (Sigma; Cat. no. 201677) w/v for 15 min after removal of TMB-ELISA by washing once with 1X TBS. Afterwards, Janus Green was aspirated followed by washing with distilled water to remove the excess stain. After washing, 800 µl of 0.5 N HCl was added to elute the stain. 200 µl of the eluate was transferred to a 96-well plate, and at 595 nm absorbance was recorded. For analysis, data were analyzed by calculating the ratio of absorbance at 450/595 followed by normalizing the value of pcDNA3.1 transfected cell reading as 1. Normalized values were plotted using GraphPad Prism v 9.5.0 software.

## NanoBiT-based βarr recruitment assay

Plasma membrane localization of βarrs upon stimulation of GPR109A with respective ligands was measured by luminescence-based enzyme-linked complement assay (NanoBiT-based assay) following the protocol described earlier[43,55]. Briefly, a receptor harboring SmBiT at the carboxy-terminus (3.5 µg) and βarr1/2 constructs (3.5 µg) with N-terminally fused LgBiT were co-transfected in HEK-293 cells using the transfection reagent polyethyleneimine (PEI) linear at DNA: PEI

ratio of 1:3. Moreover, for βarr recruitment downstream of GPR109A mutants, bystander approach of βarr recruitment has been employed. For the NanoBiT-based bystander βarr recruitment, cells were transfected with 3 µg of mentioned receptor constructs, 2 µg of βarr1 fused with SmBiT at the N-terminus and 5 µg of N-terminus LgBiT fused plasma membrane localizing tag, CAAX motif derived from human K-RAS protein. Post 16–18 h of transfection, cells were trypsinized, and resuspended in the NanoBiT assay buffer containing 1X HBSS, 0.01% BSA, 5 mM HEPES pH 7.4, and 10 µM coelenterazine (GoldBio, Cat. no. CZ05). Cells were then seeded in opaque flat bottom white 96 well plate at a density of $1 \times 10^5$ cells well$^{-1}$ and incubated for 120 min (90 min at 37 °C, followed by 30 min at room temperature). Post incubation, basal level luminescence readings were taken, followed by ligand addition. A series of ligand concentrations were prepared using a buffer solution composed of 1X HBSS and 5 mM HEPES at pH 7.4. Subsequently, cells were stimulated with different doses of the specified ligands. Luminescence upon stimulation was recorded up to 20 cycles by a multi-mode plate reader. For analysis, stimulated readings were normalized with respect to the signal of minimal ligand concentration as 1 and plotted using nonlinear regression analysis in GraphPad Prism v 9.5.0 software.

## NanoBiT-based G-protein dissociation assay

Agonist-induced G-protein activation was measured by a NanoBiT-based G-protein dissociation assay described previously[43,55]. Briefly, HEK-293 cells were transfected with 1 µg of LgBiT-tagged Gα subunit, 4 µg of SmBiT-tagged Gγ2 subunit, 4 µg of untagged Gβ1 subunit along with 1 µg of untagged receptor construct using transfection reagent PEI at DNA: PEI ratio of 1:3. Post transfection, cells were harvested and seeded in a 96 well plate at a density of $1 \times 10^5$ cells well$^{-1}$. Cells were seeded in buffer containing 1X HBSS, 0.01% BSA, 5 mM HEPES pH 7.4, and 10 µM coelenterazine and incubated for 120 min (90 min at 37 °C and 30 min at room temperature). Post incubation, 3 cycles of basal level luminescence readings were recorded using a multi-mode plate reader. After that, cells were stimulated with varying ligand concentrations ranging from 10 pM to 10 µM. After stimulation, 20 cycles of luminescence were recorded. For data analysis, values after 15 min of stimulation were used and normalized with respect to the signal at the minimal ligand concentration as 1. Normalized values were plotted using nonlinear regression analysis in GraphPad Prism v 9.5.0 software.

## Bias factor calculation

Bias factor (β) for transducer activation was measured using the software https://biasedcalculator.shinyapps.io/calc/. E$_{max}$ and EC50 values from the compiled data of desired experiments were used to calculate bias factor and the value was plotted using GraphPad Prism software (9.5.0). Formula used for the bias factor (β ± Δβ) calculation is mentioned below.

$$RA = \frac{E_{max,P1} \cdot EC50_{P2}}{EC50_{P1} \cdot E_{max,P2}}$$

$$\beta = \log_{10}\left(\frac{RA_{lig}}{RA_{ref}}\right)$$

$$\Delta RA = RA \cdot \sqrt{\left(\frac{\Delta E_{max,P1}}{E_{max,P1}}\right)^2 + \left(\frac{\Delta EC50_{P1}}{EC50_{P1}}\right)^2 + \left(\frac{\Delta E_{max,P2}}{E_{max,P2}}\right)^2 + \left(\frac{\Delta EC50_{P2}}{EC50_{P2}}\right)^2}$$

$$\Delta\beta = 0.434\sqrt{\left(\frac{\Delta RA_{lig}}{RA_{lig}}\right)^2 + \left(\frac{\Delta RA_{ref}}{RA_{ref}}\right)^2}$$

## Data quantification and statistical analysis

All the experiments described in the current study were performed at least three times. The data (mean ± SEM) are plotted and analyzed using GraphPad Prism software (9.5.0). For plotting data were normalized with respect to proper experimental controls as mentioned in the corresponding figure legends.

## Reporting summary

Further information on research design is available in the Nature Portfolio Reporting Summary linked to this article.

## Data availability

The data that support this study are available from the corresponding authors upon request. All the data are included in the manuscript and any additional information required to reanalyze the data reported in this paper is available from the corresponding author upon reasonable request. The cryo-EM maps and structures have been deposited in the PDB and EMDB with accession numbers 8IY9 and EMD-35817 for niacin-GPR109A-Go, 8JER and EMD-36193 for acipimox-GPR109A-Go, 8IYW and EMD-35831 for GSK256073-GPR109A-Go, 8IYH and EMD-35822 for MK6892-GPR109A-Go and EMD-36280, PDB ID: 8JHN for MMF-GPR109A-Go complex. The accession codes of other PDB coordinate files referenced in this study are 8HPT for C5a-pep bound mouse C5aR1 and 7ZL9 for the inactive crystal structure of GPR109A. Source data are provided with this paper.

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

## Acknowledgements
Research in A.K.S.'s laboratory is supported by the Senior Fellowship of the DBT Wellcome Trust India Alliance (IA/S/20/1/504916) awarded to A.K.S., Science and Engineering Research Board (IPA/2020/000405), Young Scientist Award from Lady Tata Memorial Trust, and IIT Kanpur. Cryo-EM was performed at BioEM lab of the Biozentrum at the University of Basel, and we thank Carola Alampi and David Kalbermatter for their excellent technical assistance.

## Author contributions
MKY expressed and purified GPR109A, and reconstituted the receptor-G-protein complexes with help from VS and GM; PS carried out the pharmacological and cellular assays on GPR109A with help from SM, AD and NZ; S. Saha purified mini-Gαo and ScFv16 with help from S. Sharma, RB performed negative-staining EM, processed the cryo-EM data, prepared and deposited the coordinates with help from JM, and drafted the figures together with MG; MC screened the samples and collected cryo-EM data; AKS supervised and managed the overall project; all authors contributed to data analysis, interpretation and manuscript writing.

## Competing interests
The authors declare no competing interests.
