## [Peer Review File · Nature Communications]

Structure-guided engineering of biased-agonism in the human niacin receptor via single amino acid substitutionReviewer #1 (Remarks to the Author):

I have read the manuscript by Yadav et al. with great interest. This manuscript provides a molecular understanding of GPR109A- G-protein activation and coupling in the presence of different agonists. The authors have elucidated 5 structures of the receptor coupled to the G-protein in the presence of different agonists with unique scaffolds and chemical properties especially relating to their pharmacological effects and clinical use. The study is extensive and thorough with structural analysis supported by functional and mutagenesis studies. In my opinion, this work is certainly of general importance and a perfect fit for Nat. Comms.

I have a list of comments and suggestions which I request the author to pay attention to and address:

1. In your author affiliations, the second affiliation (Basel, Switzerland) has not been assigned to any of the authors, are you missing an author?

2. Figure legends 1 a and 1 b seem to have been interchanged, kindly rectify.

3. I believe that plots in Fig 1c-e can be merged into a single plot, that would enable a better comparative analysis between the different agonists instead of individually comparing them to Niacin.

4. Why calculate bias factor only for MK6892? Why not show it for all the ligands of the study (Fig 1f). You may add them to the supplementary information if not in the main text.

5. Figure 2 can be improved greatly.

a. The overall resolution of each map can be added to the figure.

b. The color scheme is confusing. For example, the color assigned to Ga in 2a is then used for the receptor in 2b. Similarly, grey is used for Ga subunit in 2b but then for the receptor in 2c.

c. In the bottom half of the panel (ribbon representation) I do not see the ligands at all except in 2c.

d. The ligand representation in stick for acipimox (2b) looks orange (similar to 2d) but then in fig 3, it is depicted in red.

6. Fig 3a. 'Superimposed structure of ligand bound GPR109A', however the ligand is not shown.

7. In fig 3h, if you use an alternate font color for R111 and S178/ S179; it may further highlight their importance as the rest of your work specifically focuses on these two residues.

8. Throughout the manuscript, amino acid residues are interchangeably referred by their three letter and single letter annotations. Please select and use either one. For example: Arg111 on line 172 and 250 but R111 on line 253. Same is true for Ser179, S179 and so on.

9. Again, in fig 5 (especially c, i), the ligands are not visible at all. I understand that the focus of that figure is on Ga docking on the receptor but ensuring that the ligands are visible would be important. Maybe changing your color scheme will do the trick.

10. The authors have missed referencing at a few important parts of the manuscript. Line 129, 160, 177, etc.

11. Line 205 "recently determined inactive state crystal structure of GPR109A", kindly provide accession id and reference. Also lines 280-282 " As mentioned earlier,... is responsible for the lipid lowering effect" should be referenced.

12. I also suggest that at some point in the introduction, stating niacin is vitamin B3 may benefit early career readers.

13. I request the authors to consider rewording most of their discussion. Words "slightly more", "slightly weaker", "tempting" can be completely avoided and actual statistics can be mentioned. Lines 286-288 " Our mutagenesis studies guided....yield a mutant"; please mention the mutant. Line 296 " the experimental framework....should " should can be reworded to 'may potentially or could'.

14. GPR109A purification method: " The purified receptor was treated with two rounds of 2 mM Iodoacetamide, followed by one round of 2 mM cysteine treatment to remove free Iodoacetamide." Please elaborate on treated. Was there any washing/ column exchange/ desalting involved or just incubation.

15. Cryo -EM sample preparation: kindly mention final protein concentrations and blotting times.

16. Data processing: kindly elaborate on 'auto picking'. What type of picking was used: Template based or template free? If a template was used, what was the ref structure. Or if it was blob picking: was it circular/ ring/ elliptical and the size?

17. For all transfections: GloSensor-based cAMP assay, NanoBiT-based β arr recruitment and G-protein dissociation assays, Transfection volumes and cell densities are missing. Without knowing the volume and cell count, sharing the amount of DNA used makes little sense. Kindly add a more detailed protocol.

18. MD-simulation analysis on the mutations could further provide insights to receptor inactivation and biased-agonism understanding. However, its absence is not a major limitation and I leave it to the authors if they would consider adding it to this study.

Reviewer #2 (Remarks to the Author):

This manuscript by Yadav and Sarma et al. aims to extend our understanding of the HCA2 receptor by presenting several cryo-EM structures bound to various ligands and the two engineered mutants. Despite recent similar contributions by Yang et al. and others, the authors claim to provide a previously lacking structural framework for understanding the receptor.

Although the current paper goes a step further by introducing additional ligands and engineering receptor mutants to manipulate agonist-induced coupling or bias, the structural framework is not lacking, as stated by the authors. Moreover, the R111A and adjacent S178A receptor mutants were tested previously in terms of cAMP response and Gi coupling (Yang et al., 2023). This was not sufficiently acknowledged in the manuscript.

Moreover, the structures described here are of a significantly lower resolution and quality compared to the recently reported ones (Yang et al., Nature Communications, 2023; Zhao et al., Molecular Cell, 2023; and Suzuki et al. (available in PDB)). The manuscript does not seem to provide novel or significant advances in understanding receptor activation and signaling.

Additionally, the manuscript has major flaws related to cryo-EM data and structure interpretation. Specifically:

1. Upon checking the cryo-EM data, repetitive issues with the data processing were observed. Representative raw micrographs for four out of five datasets have vertical and horizontal stripes, which are artifacts appearing due to incorrect processing of the datasets (see Supplementary figures S4a, S5a, S6a, S7a). A reason for that could be the gain correction of the micrographs. In this case, a gain reference could be estimated from the raw data and used as a new reference for alignment. These datasets must be reprocessed with correct gain referencing.
2. While these datasets require reprocessing, there are general concerns about the maps and models.

Despite the moderate resolution, authors could build in the small molecule ligands in the density. However, there are multiple problems with that. In the MMF-bound structure, no pronounced density is visible for the ligand in the provided map. It was built into the noise, not to mention the geometry issues. The GSK256073-bound structure has major clashes in the ligand binding pocket, where the ligand clashes with F180 and S179 of the receptor in addition to distorted bond angles and lengths of the ligand. See PDB reports, which indicate the geometry issues as well. The MK6892-bound structure exhibits similar problems with ligand geometry, including incorrect torsion angles. The other models must be validated and fixed in a similar manner. Coot, used for the model building in the current study, has all the necessary functionality for detecting these issues under the ligand tab.

3. The model building for certain segments appears to be based on noise rather than on actual density. For example, in the acipimox-GPR109A-G protein complex, while the G protein has reasonable density, the receptor has a lower signal-to-noise ratio, and some of the helices do not have density supporting the building of the side chains. One needs to go up in contour levels to see the density of helices. The upper part of the receptor is overfitted and cannot be built in confidently. This yields misleading interpretations; parts of the models must be removed, and all the models require reevaluation.

4. Lines 136-137: "Still however, in each of these complexes, the ligand densities were clearly discernible, allowing us to visualize ligand-receptor interactions, ..." . This statement is inaccurate, as noted in the comments above regarding ligand density (point 3).

5. Lines 145-160: The claims are not correct. The density of the extracellular lid is overfitted and much better defined in the already reported structures (Yang et al., 2023; Zhou et al., 2023).

6. The agonist-induced activation section should be rewritten in the retrospective of the previously reported structures by Yang et al. and potentially the Zhao paper as well. The section should address what we learn new about the receptor activation mechanism from the structures reported here if we learn anything new.

7. The same applies to the "The interface of GPR109A-G-protein interaction" section: How does it compare to the Gi-bound structures? What is new in Go-coupled structures?

Some minor points are listed below:

1. There is no hyphen between G and protein. Correct across the manuscript and figure legends.
2. Several references are missing along the text, indicated as REF by the authors (lines 129, 160, 177)
3. Line 129: The term "state-of-the-art methodology" seems to overstate the case, as the methods described are standard practices within the field.
4. Line 133: "coulombic maps" term is not commonly used to describe cryo-EM maps since it can be confused with electrostatic potential maps. It is better to change it to a cryo-EM map or a similar term.
5. Line 261: EC50 2.59±125 nM should be EC50 2.59±1.25 nM
6. Line 253: Use one letter code for amino acids for consistency.
7. GPR109A purification: Specify which amino acids of the M4R N-termini were taken.
8. NanoBiT-based recruitment assay. Redundancy in citations of the protocols leads to unnecessary self-citation (line 585). The NanoBiT-based assay was described by Inoue et al., 2019 (which must be cited for G protein assay as well instead or in addition to 33). It was later adapted for arrestin recruitment assay. The citation 45 is unnecessary here. Removing this citation and adding the relevant citation is highly recommended.
9. Supplementary Figure 10: The densities seem to be displayed at the different contour levels for different regions. Indicate the level. Moreover, the term "electron density maps" is simply wrong. X-rays produce electron density maps, whereas electrons produce density maps. Correct to density maps.
10. Figure 1c, third graph from the top on the left side (cAMP response): The curve does not seem to follow the data points for acipimox.
11. Figures are overloaded with panels and often fail to convey the main message. Some panels are redundant, e.g., Figure 3 has three times the superposition of the receptor backbone (panels a,f, and i). It could be reduced to one superposition, while the other panels could only show the zoomed regions of interest.

Reference: NCOMMS-23-35094

Response to reviewers' comments

Reviewer #1

I have read the manuscript by Yadav et al. with great interest. This manuscript provides a molecular understanding of GPR109A-G-protein activation and coupling in the presence of different agonists. The authors have elucidated 5 structures of the receptor coupled to the G-protein in the presence of different agonists with unique scaffolds and chemical properties especially relating to their pharmacological effects and clinical use. The study is extensive and thorough with structural analysis supported by functional and mutagenesis studies. In my opinion, this work is certainly of general importance and a perfect fit for Nat. Comms.

We thank the reviewer for her/his positive and insightful comments on our manuscript. We have now addressed the specific comments as outlined below.

I have a list of comments and suggestions which I request the author to pay attention to and address:

1. In your author affiliations, the second affiliation (Basel, Switzerland) has not been assigned to any of the authors, are you missing an author?

We have corrected this in the revised manuscript.

2. Figure legends 1 a and 1 b seem to have been interchanged, kindly rectify.

We have corrected this in the revised manuscript.

3. I believe that plots in Fig 1c-e can be merged into a single plot, that would enable a better comparative analysis between the different agonists instead of individually comparing them to Niacin.

We thank the reviewer for this suggestion. These experiments were carried out independent of each other with niacin as a reference, and therefore, merging them in one figure would not be appropriate. However, following reviewer's suggestion, we have now presented the key results from these experiments (i.e., E_{max} and EC_{50}) in a tabular format (Supplementary Fig. 2a).

4. Why calculate bias factor only for MK6892? Why not show it for all the ligands of the study (Fig 1f). You may add them to the supplementary information if not in the main text.

Following reviewer's suggestion, we have now calculated the bias factor for all the ligands tested here, and present this information in Table S2B.

5. Figure 2 can be improved greatly.

a. The overall resolution of each map can be added to the figure.

Following reviewer's suggestion, we have now included this information in the revised Fig. 3a-e.

b. The color scheme is confusing. For example, the color assigned to G α in 2a is then used for the receptor in 2b. Similarly, grey is used for G α subunit in 2b but then for the receptor in 2c.

We thank the reviewer for this excellent suggestion, and following her/his advice, we have now unified the color scheme of the common components i.e. G-proteins and ScFv16, and used different colors for the receptor and ligands (please refer to the revised Fig. 3a-e).

c. In the bottom half of the panel (ribbon representation) I do not see the ligands at all except in 2c.

Following reviewer's excellent suggestion, we have revised the color scheme in the Figure to represent the ligands better, and indicated them with corresponding labels (please refer to the revised Fig. 3a-e). We also note that some of these ligands such as niacin, acipimox, and MMF are too small, and therefore, do not reflect well in the ribbon diagram.

d. The ligand representation in stick for acipimox (2b) looks orange (similar to 2d) but then in fig 3, it is depicted in red.

We have now corrected this in the revised manuscript (Fig. 3b).

6. Fig 3a. 'Superimposed structure of ligand bound GPR109A', however the ligand is not shown.

We have now included all the ligands in superimposed image and also shown superimposed ligands in inset (Fig. 4c).

7. In fig 3h, if you use an alternate font color for R111 and S178/ S179; it may further highlight their importance as the rest of your work specifically focuses on these two residues.

Following reviewer's excellent suggestion, we have now represented these two residues in different colors to highlight their importance (revised Fig. 5a).

8. Throughout the manuscript, amino acid residues are interchangeably referred by their three letter and single letter annotations. Please select and use either one. For example: Arg111 on line 172 and 250 but R111 on line 253. Same is true for Ser179, S179 and so on.

We thank the reviewer for this excellent suggestion, and we have now carefully addressed this in the revised manuscript. We have used single letter annotations for amino acids throughout the manuscript.

9. Again, in fig 5 (especially c, i), the ligands are not visible at all. I understand that the focus of that figure is on G α docking on the receptor but ensuring that the ligands are visible would be important. Maybe changing your color scheme will do the trick.

Following reviewer's excellent suggestion, we have now adjusted the color scheme of the ligands to highlight them, and have also indicated with corresponding labels (Fig. 7).

10. The authors have missed referencing at a few important parts of the manuscript. Line 129, 160, 177, etc.

We thank the reviewer for pointing this out, and we have now included the missing references in the revised manuscript.

11. Line 205 "recently determined inactive state crystal structure of GPR109A", kindly provide accession id and reference. Also lines 280-282 " As mentioned earlier,... is responsible for the lipid lowering effect" should be referenced.

We thank the reviewer for these excellent suggestions, and we have now addressed this in the revised manuscript (references 29 and 10, line 202 and 284, respectively).

12. I also suggest that at some point in the introduction, stating niacin is vitamin B3 may benefit early career readers.

We have now mentioned that niacin is also known as Vitamin B3 in the introduction (line 67, page 3) in the revised manuscript.

13. I request the authors to consider rewording most of their discussion. Words "slightly more", "slightly weaker", "tempting" can be completely avoided and actual statistics can be mentioned.

We thank the reviewer for this suggestion, and we have now addressed this in the revised manuscript.

Lines 286-288 "Our mutagenesis studies guided....yield a mutant", please mention the mutant.

We have now mentioned the mutants in the revised manuscript (line 287-288, page 10).

Line 296 " the experimental framework....should " should can be reworded to 'may potentially or could'.

We have corrected this in the revised manuscript (line 298, page 11).

14. GPR109A purification method: “ The purified receptor was treated with two rounds of 2 mM Iodoacetamide, followed by one round of 2 mM cysteine treatment to remove free Iodoacetamide.” Please elaborate on treated. Was there any washing/ column exchange/ desalting involved or just incubation.

We typically add iodoacetamide to the purified receptor at 2mM final concentration, and incubate it for 30min on ice. Afterwards, we add 2mM L-cysteine and incubate it for an additional 30min. We have now revised the corresponding method section to reflect this in the revised manuscript (line 357-358, page 13).

15. Cryo-EM sample preparation: kindly mention final protein concentrations and blotting times.

We have now mentioned the final protein concentration and blotting time for different complexes in the corresponding section in the revised manuscript (line 437, page 16).

16. Data processing: kindly elaborate on ‘auto picking’. What type of picking was used: Template based or template free? If a template was used, what was the ref structure. Or if it was blob picking: was it circular/ ring/ elliptical and the size?

We thank the reviewer for pointing this out, and we have now provided the corresponding information in the revised manuscript (line 449-450, page 16). Briefly, the blob-picker (template free) sub-program was used to automatically pick particles with the circular blob in the diameter range of 100-220 Å.

17. For all transfections: GloSensor-based cAMP assay, NanoBiT-based β arr recruitment and G-protein dissociation assays, Transfection volumes and cell densities are missing. Without knowing the volume and cell count, sharing the amount of DNA used makes little sense. Kindly add a more detailed protocol.

We thank the reviewer for pointing this out, and we have now provided the corresponding information in the revised manuscript (line 524-531, page 19).

18. MD-simulation analysis on the mutations could further provide insights to receptor inactivation and biased-agonism understanding. However, its absence is not a major limitation and I leave it to the authors if they would consider adding it to this study.

We thank the reviewer for this interesting suggestion. We hope to employ MD simulation to probe receptor activation and biased-agonism in future studies.

Reviewer #2:

This manuscript by Yadav and Sarma et al. aims to extend our understanding of the HCA2 receptor by presenting several cryo-EM structures bound to various ligands and the two engineered mutants. Despite recent contributions by Yang et al. and others, the authors claim to provide a previously lacking structural framework for understanding the receptor.

We thank the reviewer for her/his time to go through our manuscript carefully, and providing overall positive feedback.

Although the current paper goes a step further by introducing additional ligands and engineering receptor mutants to manipulate agonist-induced coupling or bias, the structural framework is not lacking, as stated by the authors. Moreover, the R111A and adjacent S178A receptor mutants were tested previously in terms of cAMP response and Gi coupling (Yang et al., 2023). This was not sufficiently acknowledged in the manuscript.

We understand and appreciate the point mentioned by the reviewer, and we are also aware of the other papers that have appeared while our manuscript was under review. We also note that our manuscript was posted in bioRxiv earlier this year, and we believe that it is basically contemporary to other studies that have appeared recently. We have now cited and discussed the paper by Yang et al., and other relevant papers that have appeared in the meantime. As the reviewer correctly points out, Yang et al., have measured Gi-coupling for R111A and S178A mutants, however, we present a more comprehensive transducer-coupling profile of R111A and S179A mutants to draw important insights from the perspective of biased-agonism (Fig. 7 in the revised manuscript).

Moreover, the structures described here are of a significantly lower resolution and quality compared to the recently reported ones (Yang et al., Nature Communications, 2023; Zhao et al., Molecular Cell, 2023; and Suzuki et al. (available in PDB). The manuscript does not seem to provide novel or significant advances in understanding receptor activation and signaling.

We understand the point mentioned by the reviewer regarding the resolution, however, we disagree that our structures are of lower quality than the other structures. As presented below in Figure R2.1 and Fig. R2.2., the overall maps of the structures presented in the present manuscript are comparable to other structures that have just become available in PDB. As mentioned above, we believe that the current study contemporary to other studies that have appeared recently. Moreover, we present a far more comprehensive transducer-coupling profile of receptor mutants designed based on structural insights than any of the other manuscripts, which appeared while the current manuscript was under review.

Figure R2.1. Comparison of the overall structural maps determined in the current study vs. highest-resolution structures that have appeared while the current study was under review. Acipimox-GPR109A-G-protein structures is not compared as the corresponding entry from other study has not been released yet.

Niacin-GPR109A-G-protein		
Ramachandran Plot	Current study	8JIL
Favored (%)	96.94	95.03
Allowed (%)	3.06	4.97
Disallowed (%)	0	0
Molprobity Score	1.38	1.98
Clash Score	3.88	13.36
MK6892-GPR109A-G-protein		
Ramachandran Plot	This study	8IHF
Favored (%)	97.73	96.14
Allowed (%)	2.27	3.86
Disallowed (%)	0	0
Molprobity Score	1.35	1.59
Clash Score	5.05	5.98
GSK256073-GPR109A-G-protein		
Ramachandran Plot	This study	8IHB
Favored (%)	96.73	96.50
Allowed (%)	3.27	3.50
Disallowed (%)	0	0
Molprobity Score	1.62	1.59
Clash Score	5.43	6.50
MMF-GPR109A-G-protein		
Ramachandran Plot	This study	8JIM
Favored (%)	96.63	94.16
Allowed (%)	3.37	5.75
Disallowed (%)	0	0
Molprobity Score	1.55	1.93
Clash Score	6.14	10.38

Figure R2.2. Comparison of the Ramachandran statistics for the structures determined in the current study vs. highest-resolution structures that have appeared while the current study was under review. Acipimox-GPR109A-G-protein structures is not compared as the corresponding entry from other study has not been released yet.

1. Upon checking the cryo-EM data, repetitive issues with the data processing were observed. Representative raw micrographs for four out of five datasets have vertical and horizontal stripes, which are artifacts appearing due to incorrect processing of the datasets (see Supplementary figures S4a, S5a, S6a, S7a). A reason for that could be the gain correction of the micrographs. In this case, a gain reference could be estimated from the raw data and used as a new reference for alignment. These datasets must be reprocessed with correct gain referencing.

We have processed all the cryo-EM data only after appropriate gain correction and referencing. However, we inadvertently included the representative raw images in the data processing pipeline before gain referencing that leads to the appearance of vertical and

horizontal stripes. We thank the reviewer for pointing this out, and we have now corrected this in the revised manuscript (Supplementary Fig. 5-9) (included below in Figure R2.3).

2. While these datasets require reprocessing, there are general concerns about the maps and models. Despite the moderate resolution, authors could build in the small molecule ligands in the density. However, there are multiple problems with that. In the MMF-bound structure, no pronounced density is visible for the ligand in the provided map. It was built into the noise, not to mention the geometry issues. The GSK256073-bound structure has major clashes in the ligand binding pocket, where the ligand clashes with F180 and S179 of the receptor in addition to distorted bond angles and lengths of the ligand. See PDB reports, which indicate the geometry issues as well. The MK6892-bound structure exhibits similar problems with ligand geometry, including incorrect torsion angles. The other models must be validated and fixed in a similar manner. Coot, used for the model building in the current study, has all the necessary functionality for detecting these issues under the ligand tab.

We understand the reviewer's point about moderate resolution; however, we respectfully disagree with the comment about ligand density. For example, as shown below in Figure R2.4, the density for MMF is apparent, albeit partial, even at different contour levels. Still however, we have now mentioned this point in the main text (line 136-137, page 5). Moreover, following reviewer's suggestion, we have now further refined the structures to improve the overall models and geometry constraints as presented below in Figure R2.5. These changes are now reflected in the revised PDB validation reports as well, which are uploaded with the manuscript files. We also note that overall positioning of the ligands in our structures are identical to those observed in recently reported higher resolution structures (Figure R2.5).

Niacin-GPR109A-G-protein		
	Previous	Revised
Bond length ($ Z > 2$)	2	0
Bond angle ($ Z > 2$)	2	0
Torsion outliers	0	0
Clashes	0	0
Acipimox-GPR109A-G-protein		
	Previous	Revised
Bond length ($ Z > 2$)	1	1
Bond angle ($ Z > 2$)	2	2
Torsion outliers	0	0
Clashes	1	1
MK6892-GPR109A-G-protein		
	Previous	Revised
Bond length ($ Z > 2$)	7	0
Bond angle ($ Z > 2$)	9	0
Torsion outliers	10	2
Clashes		0
GSK256073-GPR109A-G-protein		
	Previous	Revised
Bond length ($ Z > 2$)	5	2
Bond angle ($ Z > 2$)	6	0
Torsion outliers	1	1
Clashes	6	0
MMF-GPR109A-G-protein		
	Previous	Revised
Bond length ($ Z > 2$)	3	0
Bond angle ($ Z > 2$)	3	0
Torsion outliers	6	5
Clashes	0	0

Figure R2.4. Validation statistics of various ligands in the structures determined in the current study before and after further refinement and corrections.

3. The model building for certain segments appears to be based on noise rather than on actual density. For example, in the acipimox-GPR109A-G protein complex, while the G protein has reasonable density, the receptor has a lower signal-to-noise ratio, and some of the helices do not have density supporting the building of the side chains. One needs to go up in contour levels to see the density of helices. The upper part of the receptor is overfitted and cannot be built in confidently. This yields misleading interpretations; parts of the models must be removed, and all the models require re-evaluation.

We understand the reviewer's point, and we note that some of these aspects such as minor overfitting may have resulted from the use of a tight mask during local refinement. Therefore, following the reviewer's advice, we have now refined the structures using softer masks, and the overall density of the resulting maps has improved (Figure 2.6). Moreover, we have deleted the residues which were lacking clear densities and truncated the sidechains of those residues that exhibit poor densities to C β in the structures. These changes are now reflected in the revised PDB validation reports, which are uploaded with the manuscript files.

4. Lines 136-137: “Still however, in each of these complexes, the ligand densities were clearly discernible, allowing us to visualize ligand-receptor interactions, ...” . This statement is inaccurate, as noted in the comments above regarding ligand density (point 3).

Following reviewer’s suggestion, we have revised the corresponding sentences accordingly in the revised manuscript (line136-137, page 5).

5. Lines 145-160: The claims are not correct. The density of the extracellular lid is overfitted and much better defined in the already reported structures (Yang et al., 2023; Zhou et al., 2023).

We understand the reviewer’s point about moderate resolution. However, as presented below (Figure R2.7), the overall density of the extracellular lid in our structures are comparable to those observed in recently reported higher resolution structures.

6. The agonist-induced activation section should be rewritten in the retrospective of the previously reported structures by Yang et al. and potentially the Zhao paper as well. The section should address what we learn new about the receptor activation mechanism from the structures reported here if we learn anything new.

As mentioned earlier, the findings described in our manuscript are contemporary to other studies that have appeared recently. In addition, the overall conclusion drawn in these recent papers are in agreement with that in our study in terms ligand binding, receptor activation, and transducer-coupling. Still however, following reviewer’s advice, we have now included an additional paragraph in the discussion section to cited and highlight the relevant papers that have appeared while our study was under review (line 302-309, page 11). We have also included an additional supplemental figure in the revised manuscript to underscore this point (Supplementary Fig. 16).

7. The same applies to the “The interface of GPR109A-G-protein interaction” section: How does it compare to the Gi-bound structures? What is new in Go-coupled structures?

Following reviewer’s advice, we have compared the interface of the receptor-Go-protein interaction, and observed it to be very similar to that in receptor-Gi structures reported recently (Supplementary Fig. 16).

Some minor points are listed below:

1. There is no hyphen between G and protein. Correct across the manuscript and figure legends.

Following reviewer's advice, we have now corrected this throughout the manuscript.

2. Several references are missing along the text, indicated as REF by the authors (lines 129, 160, 177).

We thank the reviewer for pointing this out, and we have now corrected this throughout the manuscript.

3. Line 129: The term "state-of-the-art methodology" seems to overstate the case, as the methods described are standard practices within the field.

Following reviewer's advice, we have now corrected this in the revised manuscript.

4. Line 133: "coulombic maps" term is not commonly used to describe cryo-EM maps since it can be confused with electrostatic potential maps. It is better to change it to a cryo-EM map or a similar term.

Following reviewer's advice, we have now corrected this in the revised manuscript.

5. Line 261: EC50 2.59 ± 125 nM should be EC50 2.59 ± 1.25 nM

Following reviewer's advice, we have now corrected this in the revised manuscript (line 260, page 10).

6. Line 253: Use one letter code for amino acids for consistency.

Following reviewer's advice, we have now corrected this in the revised manuscript and used consistent single letter code for the amino acids throughout.

7. GPR109A purification: Specify which amino acids of the M4R N-termini were taken.

Following reviewer's advice, we have now included this information in the revised manuscript (line 339, page 16).

8. NanoBiT-based recruitment assay. Redundancy in citations of the protocols leads to unnecessary self-citation (line 585). The NanoBiT-based assay was described by Inoue et al., 2019 (which must be cited for G protein assay as well instead or in addition to 33). It was

later adapted for arrestin recruitment assay. The citation 45 is unnecessary here. Removing this citation and adding the relevant citation is highly recommended.

Following reviewer's advice, we have made the corresponding changes in the revised manuscript (new reference 49).

9. Supplementary Figure 10: The densities seem to be displayed at the different contour levels for different regions. Indicate the level. Moreover, the term "electron density maps" is simply wrong. X-rays produce electron density maps, whereas electrons produce density maps. Correct to density maps.

We thank the reviewer for the suggestion, and following her/his advice, we have now indicated the contour levels for individual densities (Supplementary Fig. 11 in the revised manuscript), and changed "electron density map" to "density map" in the figure legend and main text.

10. Figure 1c, third graph from the top on the left side (cAMP response): The curve does not seem to follow the data points for acipimox.

We thank the reviewer for the suggestion, and following her/his advice, we have now repeated this experiment and incorporated the new data in the revised manuscript. We have also included it below in Figure R2.7 for ready reference.

11. Figures are overloaded with panels and often fail to convey the main message. Some panels are redundant, e.g., Figure 3 has three times the superposition of the receptor backbone (panels a,f, and i). It could be reduced to one superposition, while the other panels could only show the zoomed regions of interest.

We understand reviewer's comment, and following reviewer's advice, we have tried to simplify the Figures in the revised manuscript. For example, we have divided Figure 1 in to two parts i.e., Fig. 1 and 2 in the revised manuscript. Similarly, we have also divided Figure 3 in to two parts i.e., Fig. 4 and 5 in the revised manuscript. However, we feel that in some Figures, a little redundancy is unavoidable to orient the readers towards the key message.

Reviewer #1 (Remarks to the Author):

I have read the revised manuscript by Yadav et al. and I thank the authors for their point-by-point response to the comments made during the first round of the review process. The authors have addressed most of my concerns raised and have made modifications to the text and figures as needed for clarification.

I do suggest the authors fix a few minor corrections as suggested below:

1. The discussion does not read well. It is too long, repetitive and misses out on simply stating out the novelty and key findings of this study. I suggest the authors make it a little more succinct and impactful.
2. Wordings such as 'As mentioned earlier' (line 279) 'also' (line 302) can be omitted.
3. In discussion line 290: 'This could be likelythat most likely are able to bind....', the word are is missing.
4. Line 312: 'Our findings should....' Kindly replace the word should with may/potentially/could/ could potentially/ etc.

I recommend the publication of the manuscript.

Reviewer #3 (Remarks to the Author):

The authors provide five cryoEM structures of the Niacin receptor in complex Go G protein heterotrimer and various ligands, which they also examine in signaling assays for Go G proteins and beta-arrestins with WT and mutant receptor constructs. The premise of the manuscript is of interest and provides some worthwhile data to the field. The result of MK6892 persisting to initiate signaling in the receptor mutant R111A while smaller ligands cannot stimulate signaling in the same mutant is very intriguing and provides a framework for the activation mechanism of this receptor. A greater study into this aspect would be particularly useful to the field, especially if combined with assessing similar compounds from chemical libraries. Moreover, if the micrographs are true representations of the data, then it appears that the collected data was of relatively good quality, and it's commendable that it arose from a 200 kV microscope. In fact, I suspect the data quality would likely allow for better quality final reconstructions if the datasets were subjected to additional processing/clean-up steps, considering the large numbers of particles contributing to each final reconstruction and the decent angular distribution of particles.

Although there are promising aspects to this manuscript, major concerns with the current form of the manuscript should be addressed.

-It is an overstatement to say the densities in the provided maps are "unambiguous" (line 132). There are regions in some of the maps that are relatively 'unambiguous,' but to use this term as a blanket statement for all five maps globally is inaccurate upon inspection of the maps. Many residues modeled (including some within the ligand binding pocket and extracellular lid) have no discernable or insufficient density for being included in the model.

Of particular note:

- > For the MMF complex: The extracellular beta-sheet lid is poorly modeled in general; it seems the residues may be out of register, and many residues should be stubbed at the C-beta. TM residues L83, D273, L80, and S178 also lack sufficient density. It is unclear why the pose was chosen for MMF when half of the molecule is modeled within weak density while a stronger density towards L83/Y87 is present that could fit the ligand instead. I would encourage the authors to use one of the handful of available ligand docking tools for cryoEM maps to aid modeling and help justify their ligand poses.
- > For the MK6892 complex: The modeling of Y87, L158, and L83 in the ligand binding site is

unsupported by cryoEM density.

> For the Acipimox complex: Residues 40-55 of GPR109 TM1 have only sparse density to model the backbone, and certainly not enough density to justify the modeled side chains in that region. The sidechain L83 of the ligand binding pocket is not within resolved cryoEM density. The backbone of GPR109, residues 57-58, is modeled out of the adjacent cryoEM density.

>These are only some of the most significant points of concern; the authors would do well to revisit their modeling and stub residues with insufficient density, especially along the 7TM bundles and loops.

-Furthermore, it is concerning that the provided PDB model and maps for the GSK256073-bound complex do not match what is shown in some figures. The provided model clearly shows the ligand in a different pose, seemingly 180 degrees flipped from what is shown in Fig. 3d (top panel) and Fig. 4d (fourth panel down). Please compare the attached screenshot of the provided model to Fig. 4d. You can see that in the provided model, the CI is "pointed" towards S179, while in Fig. 4d, it is "pointing" away from S179. Also, in Fig. 4d, residue Y87 is shown, but this residue is absent in the provided model. Worth noting is that the ligand's fit appears to be more appropriate in the provided PDB model than in the figure. However, the conclusions of the paper seem to stem from the model used in the figures. For example, residue F180 is only ~2.6Å from the modeled CI group of the ligand in the provided map, yet an anionic- π bond with this residue is not described. The discrepancy between the provided model and the model used for interpretation in the manuscript is a significant issue since assessing the ligand binding site interactions is a key premise of the paper, and the manuscript should be written in accordance with the updated model, and figures should be updated accordingly.

-Also, related to the provided map/model of the GSK256073-bound Go complex, the pose of the ligand is dramatically different from that of another deposited structure of GSK256073 (PDB:8IH8). It is possible that the deposited structure was added to the PDB after the submission of this manuscript; thus, it could be understandable that the authors have not mentioned or compared it to their own structure. However, since it exists in the public domain, the authors should now directly compare it, especially since the ligand pose dramatically contradicts that of the author's determined structure.

-Could the authors comment on the choice of GoA/GoB G protein subtypes for their structures and assays in the manuscript? It is unclear why the authors have specifically tested GoA/B, rather than other Gi family subtypes. Given that one of the article's main focuses is examining bias (between Arrestin vs G protein), it would make sense and be of interest to the general readership to examine bias between G protein subtypes that may be contributing to signaling differences. With the apparent difference in ligand pose between the authors' Go complex and the available PDB structure (PDB:8IHB) complex with Gi, an assessment of G protein subtypes seems especially appropriate.

-The structural aspects of the work are not particularly insightful, considering the currently available structures of the GPR109 in the PDB. The manuscript spends considerable space discussing the structures, while the signaling assay data is downplayed. Re-focusing the manuscript text on the functional assay results could make for a more impactful manuscript.

Other minor comments:

-Fig. 1a seems unnecessary; it doesn't add much to the manuscript and could be removed.

-The authors use an unverified online calculator for a "bias factor" between G protein and arrestin. Given that this is a webpage calculator, without DOI/citation, the authors should incorporate the equation used by the server into the methods section, so that the method used is not lost if the webpage is taken down at some point in the future.

-The color range selected for the local resolution maps is skewed to make the local resolutions appear 'better'. A more appropriate range should be chosen for each (e.g., in the niacin map, use a range of 2.5 (blue) to 4.5 (red)).

-Combining Figures 1 and 2 would make sense as the assays are the same for both, and the reader could compare all 5 compounds side-by-side.

Reference: NCOMMS-23-35094B

Response to reviewers' comments

Reviewer #1:

I have read the revised manuscript by Yadav et al. and I thank the authors for their point-by-point response to the comments made during the first round of the review process. The authors have addressed most of my concerns raised and have made modifications to the text and figures as needed for clarification.

We thank the reviewer for positive comments on our revised manuscript. We have now addressed the remaining minor comments as outlined below.

1. The discussion does not read well. It is too long, repetitive and misses out on simply stating out the novelty and key findings of this study. I suggest the authors make it a little more succinct and impactful.

Following reviewer's suggestion, we have now refocused the discussion primarily on functional aspects of the study with an emphasis on key insights derived into signaling-bias.

2. Wordings such as 'As mentioned earlier' (line 279) 'also' (line 302) can be omitted.

Following reviewer's suggestion, we have now made the corresponding changes in the revised manuscript.

3. In discussion line 290: 'This could be likelythat most likely are able to bind....', the word are is missing.

Following reviewer's suggestion, we have modified the corresponding sentence in the revised manuscript for clarity.

4. Line 312: 'Our findings should....' Kindly replace the word should with may/potentially/could/ could potentially/ etc.

Following reviewer's suggestion, we have now made the corresponding changes in the revised manuscript.

Reviewer #3:

The authors provide five cryoEM structures of the Niacin receptor in complex Go G protein heterotrimer and various ligands, which they also examine in signaling assays for Go G proteins and beta-arrestins with WT and mutant receptor constructs. The premise of the manuscript is of interest and provides some worthwhile data to the field. The result of MK6892 persisting to initiate signaling in the receptor mutant R111A while smaller ligands are unable to stimulate signaling in the same mutant is very interesting and provides a framework for the activation mechanism of this receptor. A greater study into this aspect would be particularly interesting to the field, especially if combined with the assessment of similar compounds from chemical libraries. If the micrographs are true representations of the data, then it appears that the collected data was of relatively good quality, and it's commendable that it arose from a 200 kV microscope. In fact, I suspect the data quality would likely allow for better reconstructions with further processing/clean-up of the datasets, considering the large numbers of particles contributing to each final reconstruction and the decent angular distribution of particles. That said, the maps and models are of reasonable quality to justify the authors' interpretations. There are only a handful of considerations that I suggest for improvement of the manuscript.

We thank the reviewer for positive comments on our revised manuscript. We have now addressed the remaining minor comments as outlined below.

1. Could the authors comment on the choice of GoA/GoB G protein subtypes for their structures and assays in the manuscript? It is unclear why the authors have specifically tested GoA/B, rather than other Gi family subtypes. Given that one of the article's main focuses is examining bias (between Arrestin vs G protein), it would make sense and be of interest to the general readership to examine bias between G protein subtypes that may be contributing to signaling differences.

We have used mini-Gao as a surrogate of Gai to reconstitute the ternary complexes, an approach that facilitates stable complex formation for structural analysis and has been validated extensively in multiple previous studies. Accordingly, we have used GaoA/B dissociation in the functional assays to maintain consistency between structural and functional analysis. We note however that we have also used cAMP response as a broad measure of Gai-coupling and activation. We agree with the reviewer that it would be interesting to assess coupling- and signaling-bias at the level of G-protein subtypes in future studies.

2. The structural aspects of the work are not particularly insightful, considering the currently available structures of the niacin receptor in the PDB. The manuscript spends considerable space discussing the structures, while the signaling assay data is downplayed. Re-focusing the manuscript text on the functional assay results could make for a more impactful manuscript.

We understand the point, however, we also note that our study is contemporary to the other studies reporting GPR109A structures that have appeared recently. Therefore, we believe that it is reasonable to describe the structural insights. Still however, following reviewer's suggestion, we have now refocused the introduction and discussion sections on functional aspects.

3. Fig. 1a seems unnecessary; it doesn't add much to the manuscript and could be removed.

We understand the point, however, we believe that the schematic in Figure 1a helps the readers understand the genesis of the study, and therefore, we have decided to keep it.

4. The authors use an unverified online calculator for a "bias factor" between G protein and arrestin. Given that this is a webpage calculator, without DOI/citation, the authors should incorporate the equation used by the server into the methods section so that the method used is not lost if the webpage is taken down at some point in the future.

Following reviewer's suggestion, we have now included the equation in the method section (page 22 in the revised manuscript).

5. The colour range selected for the local resolution maps is skewed to make the local resolutions appear 'better'. A more appropriate range should be chosen for each (e.g., in the niacin map, use a range of 2.5 (blue) to 4.5 (red)).

Following reviewer's suggestion, we have now changed the colour range in the local resolution maps in the revised manuscript (revised supplementary Fig. 5-9).

6. Combining Figures 1 and 2 would make sense as the assays are the same for both, and the reader could compare all 5 compounds side-by-side.

We understand the point, however, the data presented in Figures 1 and 2 were collected independently, and therefore, it would not be appropriate to combine these data together.